# Exercise-Induced Muscle–Fat Crosstalk: Molecular Mediators and Their Pharmacological Modulation for the Maintenance of Metabolic Flexibility in Aging

**DOI:** 10.3390/ph18081222

**Published:** 2025-08-19

**Authors:** Amelia Tero-Vescan, Hans Degens, Antonios Matsakas, Ruxandra Ștefănescu, Bianca Eugenia Ősz, Mark Slevin

**Affiliations:** 1Biochemistry and Chemistry of the Environmental Factors Department, Faculty of Pharmacy, George Emil Palade University of Medicine, Pharmacy, Science, and Technology of Târgu Mureș, 38th Gh. Marinescu Street, 540139 Târgu Mureş, Romania; amelia.tero-vescan@umfst.ro; 2Department of Life Sciences, Manchester Metropolitan University, Chester Street, Manchester M1 5GD, UK; h.degens@mmu.ac.uk (H.D.); a.matsakas@mmu.ac.uk (A.M.); 3Institute of Sport Science and Innovations, Lithuanian Sports University, LT 221 Kaunas, Lithuania; 4Pharmacognosy and Phytotherapy Department, Faculty of Pharmacy, George Emil Palade University of Medicine, Pharmacy, Science, and Technology of Târgu Mureș, 38th Gh. Marinescu Street, 540139 Târgu Mureş, Romania; ruxandra.stefanescu@umfst.ro; 5Pharmacology and Clinical Pharmacy Department, Faculty of Pharmacy, George Emil Palade University of Medicine, Pharmacy, Science, and Technology of Târgu Mureș, 38th Gh. Marinescu Street, 540139 Târgu Mureş, Romania; bianca.osz@umfst.ro; 6Center for Advanced Medical and Pharmaceutical Research, George Emil Palade University of Medicine, Pharmacy, Science, and Technology of Târgu Mureș, 38th Gh. Marinescu Street, 540139 Târgu Mureş, Romania

**Keywords:** myokines, adipokines, metabolic flexibility, mitochondrial biogenesis, skeletal muscle–adipose tissue crosstalk, IL-6, irisin, leptin, adiponectin

## Abstract

Regular physical activity induces a dynamic crosstalk between skeletal muscle and adipose tissue, modulating the key molecular pathways that underlie metabolic flexibility, mitochondrial function, and inflammation. This review highlights the role of myokines and adipokines—particularly IL-6, irisin, leptin, and adiponectin—in orchestrating muscle–adipose tissue communication during exercise. Exercise stimulates AMPK, PGC-1α, and SIRT1 signaling, promoting mitochondrial biogenesis, fatty acid oxidation, and autophagy, while also regulating muscle hypertrophy through the PI3K/Akt/mTOR and Wnt/β-catenin pathways. Simultaneously, adipose-derived factors like leptin and adiponectin modulate skeletal muscle metabolism via JAK/STAT3 and AdipoR1-mediated AMPK activation. Additionally, emerging exercise mimetics such as the mitochondrial-derived peptide MOTS-c and myostatin inhibitors are highlighted for their roles in increasing muscle mass, the browning of white adipose tissue, and improving systemic metabolic function. The review also addresses the role of anti-inflammatory compounds, including omega-3 polyunsaturated fatty acids and low-dose aspirin, in mitigating NF-κB and IL-6 signaling to protect mitochondrial health. The resulting metabolic flexibility, defined as the ability to efficiently switch between lipid and glucose oxidation, is enhanced through repeated exercise, counteracting age- and disease-related mitochondrial and functional decline. Together, these adaptations demonstrate the importance of inter-tissue signaling in maintaining energy homeostasis and preventing sarcopenia, obesity, and insulin resistance. Finally, here we propose a stratified treatment algorithm based on common age-related comorbidities, offering a framework for precision-based interventions that may offer a promising strategy to preserve metabolic plasticity and delay the age-associated decline in cardiometabolic health.

## 1. Introduction

Aging is accompanied by a progressive decline in physiological resilience, often manifesting as impaired metabolic flexibility, reduced skeletal muscle function, chronic low-grade inflammation, and increased susceptibility to sarcopenia, frailty, and cardiometabolic diseases [1]. These alterations significantly contribute to a reduced health span and an increased risk of morbidity in older adults. At the cellular level, aging is associated with mitochondrial dysfunction, insulin resistance, oxidative stress, and altered myokine–adipokine signaling, which collectively impair the body’s ability to adapt to energetic demands and external stressors [2,3]. Regular physical activity is widely recognized as one of the most effective non-pharmacological interventions to mitigate the deleterious effects of aging. Exercise induces a range of systemic and tissue-specific adaptations that enhance mitochondrial biogenesis, improve insulin sensitivity, modulate inflammatory pathways, and promote the favorable remodeling of skeletal muscle and adipose tissue [4,5,6]. These adaptations are mediated by a network of molecular signals, including AMP-activated protein kinase (AMPK), sirtuins (particularly SIRT1), peroxisome proliferator-activated receptors (PPARs), and peroxisome proliferator-activated receptor gamma coactivator 1-alpha (PGC-1α), which collectively modulate cellular energy metabolism, oxidative capacity, and metabolic homeostasis [7,8].

In particular, exercise stimulates the release of myokines and adipokines, bioactive molecules that coordinate inter-tissue communication between skeletal muscle, adipose tissue, the liver, and the brain [4]. Myokines such as irisin, IL-6, and myostatin, and adipokines such as adiponectin and leptin, play central roles in regulating glucose and lipid metabolism, mitochondrial function, and inflammation [9]. These molecules are increasingly recognized as critical mediators of the systemic benefits of exercise, particularly in aging populations [10].

However, a significant proportion of older individuals are unable to engage in sufficient levels of physical activity due to frailty, sarcopenia, multimorbidity, or functional impairments. This limitation has prompted a growing interest in the development of pharmacological agents that can mimic or enhance the molecular benefits of exercise, commonly referred to as “exercise mimetics” [11]. These compounds aim to activate the same intracellular signaling pathways modulated by physical activity, including AMPK, PGC-1α, and SIRT1, without requiring mechanical or contractile stimuli [8,12]. By targeting these pathways, exercise mimetics offer a promising therapeutic strategy to preserve metabolic flexibility and physical function in individuals with reduced exercise capacity [13].

This narrative review explores the molecular basis of muscle–adipose tissue crosstalk during physical activity, highlights the signaling pathways and bioactive mediators involved, and evaluates pharmacological strategies designed to mimic or support these exercise-induced adaptations. Furthermore, the potential for stratified therapeutic approaches tailored to clinical conditions commonly encountered in aging, such as T2DM, cardiovascular disease, sarcopenia, and frailty, are discussed. The integration of pharmacological and non-pharmacological strategies may ultimately offer a more effective means of extending health span and maintaining functional independence in aging populations.

## 2. Dynamic Regulation of Energy Substrate Use: Metabolic Flexibility Paradigm

Aging is associated with reduced metabolic flexibility and an increased risk of developing chronic diseases such as type 2 diabetes (T2DM), cardiovascular disease, and neurodegeneration. Metabolic flexibility refers to an organism’s ability to adapt to the use of different substrates at both the tissue and cellular levels for the most efficient generation of ATP and the use of energy reserves depending on the energy demands [14]. In this context, the CACH model (catabolic–anabolic cycling hormesis) proposes an alternation between catabolic stress periods, such as those induced by physical exercise or starvation, and anabolic recovery phases [15]. This cyclical process improves insulin sensitivity and has been suggested to promote mitochondrial biogenesis and efficiency, thereby enhancing the aerobic metabolic capacity of myocytes. This cycle also leads to an increase in the production of reactive oxygen species (ROS), which, while capable of damaging cellular components, including membrane lipids, structural and enzymatic proteins, and both nuclear and mitochondrial DNA, also play essential roles in signaling processes such as mitophagy and apoptosis in senescent cells [16]. During the anabolic phase, the activation of the mechanistic target of rapamycin (mTOR)-dependent pathways supports protein synthesis and muscle remodeling [17].

Mitochondrial dysfunction, chronic inflammation, and muscle–fat signaling defects are key contributors to this age-related metabolic decline, and the age-related decline in metabolic flexibility is one of the factors that is thought to contribute to both the loss of skeletal muscle mass (sarcopenia) and function (age-related muscle weakness) which have a negative impact on quality of life. Ectopic lipid accumulation, particularly intramyocellular lipid accumulation, contributes to lipotoxicity, creating a vicious cycle that further impairs metabolic flexibility. In older adults there are two possible major lifestyle interventions to help preserve metabolic flexibility: caloric restriction and exercise [14,18].

Exercise is a well-established modulator of metabolic homeostasis; however, many older individuals with chronic comorbidities are unable to maintain regular physical activity. Hence, this article provides a comprehensive and integrative perspective on how exercise-induced myokines, particularly interleukin-6 (IL-6), irisin, and myostatin, and their downstream molecular signaling pathways regulate metabolic flexibility through muscle–adipose tissue crosstalk. A key novelty is the detailed mechanistic insight into how these myokines influence white adipose tissue (WAT) browning, mitochondrial biogenesis, and lipid oxidation, particularly emphasizing the AMPK–PGC-1α–SIRT1 axis and its role in adaptive metabolic reprogramming. Furthermore, since metabolic inflexibility and inflammation are central to aging and chronic disease the manuscript bridges basic mechanistic pathways with translational potential, proposing the development of stratified pharmacologically guided exercise mimetics as a therapeutic strategy in aging and metabolic disease.

## 3. Muscle-Derived Signals During Acute vs. Chronic Exercise

Local hypoxia, along with mechanical stress and increased energy demand, induces the release of myokines particularly during acute exercise, thereby promoting a shift in energy metabolism in skeletal muscle and activating key metabolic sensors such as AMPK and PGC-1α. AMPK activation, triggered by an ATP/AMP ratio < 1, stimulates ATP-generating processes such as fatty acid oxidation, glycolysis, and glucose transport in muscle cells [19]. Simultaneously, it inhibits ATP-consuming processes including glycogen, lipid, and protein synthesis, as well as cell growth and proliferation [20]. Furthermore, active AMPK stimulates mitochondrial biogenesis and angiogenesis, and increases appetite at the hypothalamic level by stimulating the secretion of adiponectin, which in turn promotes glucose uptake by and fatty acid oxidation in muscle cells [21,22].

In regards to molecular signaling, PGC-1α activation during physical exercise induces mitochondrial biogenesis, whereas the postexercise recovery phase triggers autophagy and mitophagy to eliminate senescent, dysfunctional, or ROS-damaged mitochondria [23]. A study conducted on whole-body PGC-1α knockout (KO) and C57BL/6 wild-type (WT) mice subjected to acute physical exercise (run to failure on a treadmill) demonstrated that 90 min postexercise, KO animals presented a 40% reduction in mitochondrial content and a 25% decline in running performance, accompanied by severe lactic acidosis due to reduced aerobic capacity. In contrast, WT mice presented an increased expression of mitochondrial biogenesis markers (cytochrome oxidase subunit IV and mitochondrial transcription factor A), as well as autophagy-related factors (p62 and light chain 3) [23].

A comparative study conducted on trained (22 ± 3 years) and untrained (24 ± 4 years) young adults, and older trained (64 ± 3 years) and untrained (65 ± 6 years) adults revealed increases in p38 MAPK phosphorylation, PGC-1α and COXIV mRNA expression at 3 h postexercise, and elevated COXIV protein levels at 3 days postexercise in trained individuals, independent of age. These findings show that the capacity for mitochondrial adaptation to exercise is not inherently limited by chronological aging but rather is strongly influenced by training status [24]. The activation of the p38 MAPK–PGC-1α axis controls mitochondrial biogenesis, oxidative capacity, and muscle endurance, so physical inactivity, rather than aging per se, appears to be the dominant factor leading to mitochondrial dysfunction and impaired muscle plasticity in older adults [25].

Lower intensity chronic physical activity enhances mitochondrial function. This effect is mediated by sustained myokine signaling, epigenetic modifications, and the modulation of key regulatory pathways involved in energy homeostasis and inflammation [26]. A multi-omics analysis performed on mice subjected to a chronic exercise protocol (running for 6 weeks), revealed that, compared with those in the control group, epigenetic modifications significantly increased the expression of Cyp4a14 and Cyp4a10, which are involved in the metabolism of arachidonic acid and acetyl-coenzyme A and in the regulation of fatty acid degradation, collectively supporting energy homeostasis and anti-inflammatory responses during chronic physical activity [27].

Oxidative adaptations driven by PGC-1α, which is released during chronic endurance exercise, stimulates mitochondrial biogenesis and oxidative capacity by activating several signaling molecules such as AMPK, SIRT1, and p38 MAPK. The phosphorylation of AMPK or deacetylation by SIRT1 activates PGC-1α, which increases the expression of transcription factors such as nuclear respiratory factor 1 (NRF-1), NRF-2, and estrogen-related receptor alpha (ERRα), leading to the activation of mitochondrial transcription factor A (TFAM) and the transcription and replication of mitochondrial DNA (mtDNA). This cascade increases the expression of key enzymes belonging to electron transport chain complexes I to IV and complex V (ATP synthase), which are essential for cellular energy production [28].

Various mechanisms have been proposed to explain ROS production during exercise, which, although not yet fully understood, has traditionally been attributed to inefficient mitochondrial respiration; however, emerging evidence suggests that the underlying processes are far more complex [29,30]. Under high-workload conditions, mitochondria exhibit an increased electrochemical (proton) gradient across the inner mitochondrial membrane, which may initially reduce electron leakage per unit of oxygen consumed. However, because of the marked increase in total mitochondrial respiration, the absolute number of electrons escaping the electron transport chain (ETC) increases, even if the percentage of leakage is lower [31]. This paradox explains why exercising muscle generates more ROS despite more efficient coupling and supports the idea that ROS production is not only a marker of mitochondrial dysfunction, but also a regulated physiological signal, particularly involved in adaptation via redox-sensitive pathways such as PGC-1α, AMPK, and nuclear factor kappa-light-chain-enhancer of activated B cells (NF-κB) [32].

Exercise, in the context of skeletal muscle physiology, exerts distinct yet interconnected effects on cellular signaling and metabolic adaptation. Acute exercise triggers immediate responses to increased mechanical load, energy demand, and local hypoxia, primarily through the activation of AMPK and PGC-1α. AMPK acts as an energy sensor, enhancing glucose uptake, fatty acid oxidation, and glycolysis while simultaneously suppressing anabolic processes such as protein and lipid synthesis. Concurrently, PGC-1α promotes mitochondrial biogenesis, facilitating energy production and oxidative capacity. These acute changes are accompanied by autophagy and mitophagy during the postexercise recovery phase, which helps to clear damaged mitochondria. Experimental evidence from PGC-1α knockout mice revealed a sharp decline in mitochondrial content and exercise performance, confirming the importance of this axis in the short-term metabolic remodeling of skeletal muscle [23,33].

Sustained chronic exercise in contrast, induces long-term metabolic reprogramming characterized by sustained myokine signaling, epigenetic modifications, and enhanced mitochondrial function. This adaptive process involves the persistent activation of the AMPK, SIRT1, and p38 MAPK pathways, all of which converge on PGC-1α to regulate transcription factors such as NRF-1, NRF-2, and ERRα. These factors, in turn, stimulate TFAM and mitochondrial DNA transcription, leading to the increased expression of respiratory chain enzymes and improved ATP production. Multi-omics analyses and human cohort studies further show that trained individuals, regardless of age, presented increased levels of mitochondrial markers and more efficient myokine responses following exercise. Therefore, while acute exercise initiates immediate metabolic adjustments, chronic activity reinforces and expands these effects, driving lasting improvements in muscle oxidative capacity and energy homeostasis [34].

## 4. Key Myokines Regulating Muscle–Adipose Tissue Crosstalk and Metabolic Flexibility

Moreover, in addition to its role in locomotion, skeletal muscle also functions as an autocrine, endocrine, and paracrine tissue, especially during physical exercise, via the secretion of myokines [35]. Myokines released during exercise due to mechanical stress, increased energy demand, and local hypoxia play a key role in regulating metabolism and inflammation, suggesting promising targets for obesity prevention and treatment [36].

### 4.1. IL-6: Sensors and Mediators of Energy Status

Acute exercise induces the secretion of IL-6, which is the most abundantly and rapidly secreted myokine and functions as an energy sensor by optimizing the short-term allocation of energy reserves. Muscle-derived IL-6 has a half-life of approximately 5 min and can increase up to 100-fold during physical activity [37]. It has a pro-inflammatory effect when released in chronic conditions such as diabetes mellitus, rheumatoid arthritis, or metabolic syndrome, and acts as an anti-inflammatory myokine when released acutely during physical exercise [38].

Receptor-specific signaling underlies the dual effects of IL-6, which binds either to the membrane-bound IL-6 receptor (mbIL-6R), expressed on hepatocytes, monocyte, macrophages, lymphocytes, and skeletal myocytes during physical activity, or to its soluble counterpart (sIL-6R) via trans-signaling [38]. The activation of mbIL-6R leads to the phosphorylation of Janus kinase (JAK), which represents the first step in the JAK–signal transducer and activator of transcription (STAT) signaling pathway, particularly STAT3, which promotes lipolysis and inhibits lipogenesis [39]. The effects of IL-6 and its triggering mechanisms during exercise are summarized in Figure 1. The figure illustrates how muscle contraction, energy stress, reactive oxygen species (ROS) generation, and glycogen depletion lead to IL-6 production in skeletal muscle. Once released, IL-6 exerts endocrine effects on multiple target tissues, including adipose tissue, the liver, and other muscle groups, influencing substrate mobilization, mitochondrial function, and systemic energy homeostasis.

Key differences in IL-6 signaling between exercise and pathological conditions arise not from the exclusive use of one receptor form but from variations in receptor availability, tissue distribution, and signaling context. During exercise, mbIL-6R is transiently expressed or upregulated on responsive cells such as skeletal myocytes, hepatocytes, and immune cells, facilitating classical signaling that promotes anti-inflammatory and metabolic effects via JAK/STAT3 activation. In contrast, chronic inflammation is often associated with elevated levels of sIL-6R, which enables trans-signaling even in cells that do not express mbIL-6R. This prolongs IL-6 activity and shifts the response toward pro-inflammatory and catabolic pathways. Therefore, the balance between mbIL-6R and sIL-6R availability, along with signal duration and tissue context, determines whether IL-6 acts in a protective or pathogenic manner [40].

In addition, activated STATs directly stimulate lipid catabolism by increasing the expression of hormone-sensitive lipase (HSL), adipose triglyceride lipase (ATGL), and α-ketoglutarate dehydrogenase (AOX), the enzymes that convert α-ketoglutarate into succinyl-CoA in the Krebs cycle and reduce the activity of fatty acid synthase (FAS) [41].

Studies have shown that IL-6 release during a 30 min walk at 50% VO_2_ max in sedentary middle-aged men is minimal, whereas prolonged physical activity, such as 5 h at 40% VO_2_ max leads to a 19-fold increase in IL-6 levels [42,43,44]. Studies conducted on healthy adult men who were administered recombinant human IL-6 (rhIL-6) reported that circulating IL-6 concentrations reached approximately 40 pg/mL, compared with baseline levels of approximately 1 pg/mL, representing a 40-fold increase. This acute increase in IL-6 led to a twofold increase in systemic fatty acid oxidation after 60 min, and this increase in lipid metabolism persisted for up to 2 h after the infusion [45].

Furthermore, another study has shown that the intravenous administration of IL-6 at higher concentrations (>100 pg/mL, comparable to those used during strenuous exercise) stimulates lipolysis. Conversely, blocking IL-6 signaling with tocilizumab during exercise blunts this metabolic effect and prevents the reduction in fat, underscoring the key role of IL-6 in exercise-induced fat mobilization [46].

The effects of IL-6 on glucose metabolism are largely attributed to the exercise-mediated activation of AMPK, an effect that is not observed for IL-6 released in the context of T2DM [47]. This has led to the concept of “IL-6 resistance,” characterized by reduced AMPK activity in response to chronic IL-6 exposure [48]. While IL-6 released during exercise promotes insulin-independent glucose uptake by inducing the translocation of GLUT4 transporters in adipose tissue via the phosphoinositide 3-kinase (PI3K)/protein kinase B (Akt) signaling pathway, IL-6 also promotes hepatic gluconeogenesis by activating key gluconeogenic enzymes, including phosphoenolpyruvate carboxykinase (PEPCK) and glucose-6-phosphatase (G-6-Pase) [49,50]. Together, these coordinated actions of IL-6 enhance metabolic flexibility by simultaneously facilitating glucose uptake and endogenous glucose production, a process thought to be a consequence of the permissive effect of IL-6 on glucagon and cortisol secretion [51].

In addition, IL-6 released during exercise (possibly in synergy with irisin) has been shown to promote the browning of white adipose tissue (WAT) by increasing the expression of uncoupling protein 1 (UCP1) and stimulating mitochondrial biogenesis [52]. Another factor that may contribute to WAT browning is the exercise-induced release of catecholamines, which activate β-adrenergic receptors (β-ARs) on adipocytes and stimulate thermogenic gene expression. Additionally, fibroblast growth factor 21 (FGF21) is also upregulated by physical activity and may act downstream of these signals to further enhance the thermogenic program [53].

### 4.2. Irisin: Mediator of Browning and Metabolic Reprogramming

Irisin is a myokine cleaved from the fibronectin type III domain-containing protein 5 (FNDC5), a process regulated by PGC-1α during muscle contraction [54]. In sedentary overweight or obese female adolescents, a session of anaerobic high-intensity interval training (HIIT) significantly increased the skeletal muscle expression of irisin, whereas no such effect was observed following aerobic exercise [55], suggesting that irisin secretion is preferentially upregulated in response to anaerobic stimuli.

In adipose tissue, irisin promotes the conversion of energy-storing WAT into a more metabolically active beige phenotype by inducing the expression of UCP1. UCP1 leads to the uncoupling of oxidative phosphorylation from the ETC, resulting in the dissipation of energy as heat rather than its use in the formation of ATP. Multiple molecular mechanisms lead to this WAT transformation, including the activation of PGC-1α within adipocytes, the stimulation of AMPK, the upregulation of SIRT1 expression and activity, and the engagement of the p38 mitogen-activated protein kinase (p38 MAPK) and extracellular signal-regulated kinase (ERK) signaling pathways. These cascades collectively promote the transcription of thermogenic and mitochondrial genes such as UCP1, PR domain-containing 16 (PRDM16), transmembrane protein 26 (TMEM26), and CD137 or tumor necrosis factor receptor superfamily member 9 (TNFRSF9), as seen in Figure 2 [56,57].

An increased intracellular AMP/ATP ratio activates the AMPK/PGC-1α signaling pathway, which plays a central role in mitochondrial quality control. The coactivator PGC-1α promotes mitochondrial biogenesis through the activation of the transcription factors NRF-1, NRF-2, and ERR, supports mitophagy by increasing Beclin1 expression and the microtubule-associated protein 1A/1B-light chain 3 (LC3) LC3-II/LC3-I ratio, and induces FNDC5 gene expression. FNDC5 activation leads to elevated circulating levels of irisin, which promotes the browning of WAT by stimulating UCP1 expression, thereby contributing to improved energy metabolism and body composition in elderly individuals [58]. The activation of PGC-1α in adipocytes leads to increased mitochondrial biogenesis, increased fatty acid β-oxidation through the upregulation of key enzymes such as carnitine palmitoyltransferase 1 (CPT1) and acyl-CoA oxidase 1 (ACOX1), and increased oxygen consumption, collectively promoting a more oxidative and energy-dissipating phenotype [59].

An in vitro experiment conducted on adipocyte cultures derived from male Sprague–Dawley rats demonstrated that irisin activates UCP1 expression through the phosphorylation of p38 MAPK and ERK, signaling pathways known to mediate cellular thermogenic responses [56]. Additionally, Zhang, Y et al. (2014), stimulated RAW 264.7 macrophages with lipopolysaccharide (LPS), and showed that irisin (0–100 nM) significantly reduced the activation of the Toll-like receptor 4 (TLR4) MyD88, concomitantly reducing the phosphorylation of nuclear factor-κB (NF-κB). This subsequently led to a significant reduction in the release of pro-inflammatory cytokines, including interleukin-1β (IL-1β), tumor necrosis factor-α (TNFα), IL-6, keratinocyte chemoattractant (KC), and monocyte chemoattractant protein-1 (MCP-1), suggesting an anti-inflammatory role for irisin in macrophage-mediated immune responses [56].

In diabetic C57BL/6 mice treated with irisin (0.5 mg/kg/day) for two weeks, the ex vivo incubation of diabetic aortic segments with irisin (1 μg/mL) significantly reduced the levels of oxidative and nitrative stress markers. These effects are mediated via the inhibition of the protein kinase C beta (PKC-β)/NADPH oxidase and NF-κB/inducible nitric oxide synthase (iNOS) pathways, resulting in improved endothelial function [60]. Additionally, irisin has been reported to modulate glucose metabolism, particularly in skeletal muscle and adipose tissue, by increasing insulin sensitivity through the activation of the PI3K/Akt signaling pathway. A study by Herouvi et al. (2025), involving 77 obese prepubertal children aged 4 to 12 years, reported elevated serum irisin levels and potential irisin resistance in obese participants compared with non-obese controls. These findings suggest a compensatory mechanism in which adipocyte-derived irisin aims to stimulate brown adipose tissue activity, increase energy expenditure, and counteract insulin resistance [60,61].

### 4.3. Myostatin: Brake of Muscle–Fat Metabolic Plasticity

Myostatin is a member of the transforming growth factor-beta (TGF-β) superfamily with an inhibitory role in skeletal muscle growth. It also exerts a significant regulatory influence on adipose tissue function, energy metabolism, and systemic energy homeostasis [62].

Myostatin secretion is inversely proportional to BMI and has been shown to reduce glucose uptake, insulin resistance, and metabolic dysregulation by inhibiting muscle mass development and promoting adipose tissue accumulation. These findings position myostatin as a potential biomarker and therapeutic target in obesity-related insulin resistance, particularly in young men [63]. Although myostatin levels increase with age in both men and women, older women tend to exhibit higher serum concentrations than age-matched men do. This may contribute to a greater predisposition to sarcopenia in elderly females [64].

Skeletal muscle and adipose tissue originate from the same mesenchymal stem cells and it is therefore no surprise that myostatin plays a role in controlling the adipogenesis-to-myogenesis ratio by influencing stem cell differentiation toward the adipocyte lineage [65]. In murine models with myostatin knockout (KO), this regulatory mechanism is disrupted, resulting in reduced adipogenesis and significantly increased muscle mass. Owing to their muscle-enhancing potential, myostatin inhibitors are classified as prohibited substances under section S4 (Hormone and Metabolic Modulators) of the WADA Prohibited List [66,67].

Myostatin plays a pivotal role in adipose tissue plasticity by modulating the browning of WAT through the AMPK-PGC-1α-FNDC5 axis, a process wherein white adipocytes acquire the characteristics of brown adipocytes, including increased mitochondrial content and the expression of UCP1, thereby increasing thermogenesis and energy expenditure [68,69]. In MSTN-KO (Mstn^−/−^) mice, brown adipose tissue (BAT) signature genes such as UCP1 and PGC-1α, along with the beige adipocyte markers TMEM26 and CD137 are notably upregulated in WAT. This browning phenotype is mediated through a noncell-autonomous mechanism involving the myokine irisin (FNDC5), which is secreted from skeletal muscle. The absence of myostatin leads to the increased expression and phosphorylation of AMPK in muscle tissue, subsequently activating PGC-1α and FNDC5. Secreted FNDC5 then promotes the browning of WAT, establishing muscle–fat crosstalk that enhances energy expenditure and improves metabolic profiles [70].

In adipose tissue, myostatin promotes the expression of genes involved in lipid biosynthesis and triglyceride storage. Specifically, by activating stearoyl-CoA desaturase 5 (SCD5), it catalyzes the Δ9 desaturation of FAs, such as the conversion of palmitic acid to palmitoleic acid and of stearic acid to oleic acid [71]. Conversely, myostatin deficiency enhances the activity of lipolytic enzymes such as CPT1 and CPT2, thereby stimulating the β-oxidation of FAs in peripheral tissues and reducing lipid accumulation [71]. In male myostatin-deficient mice fed a high-fat diet (HFD), this metabolic shift has been associated with decreased serum cholesterol and triglyceride levels, a 40-fold reduction in TNF-α, and improved insulin sensitivity [72]. Similarly, in myostatin KO mice, mitochondrial bioenergetics are significantly altered, characterized by the reduced activity of key enzymes involved in the tricarboxylic acid (TCA) cycle, decreased ETC function, and impaired oxidative phosphorylation [73]. Despite these deficits, myostatin deficiency enhances basal oxygen consumption, reduces lipid peroxidation, and upregulates the glutathione antioxidant system. Additionally, it alters cardiolipin composition within the mitochondrial membrane, changes that can be partially reversed through endurance training [74].

Although myostatin KO (Mstn^−/−^) mice exhibit marked skeletal muscle hypertrophy and retain overall normal fiber morphology and contractile function, recent studies have revealed that this anabolic phenotype is accompanied by mitochondrial dysfunction, including impaired oxidative phosphorylation, altered cardiolipin composition, and the reduced activity of the tricarboxylic acid (TCA) cycle and ETC enzymes [73,74]. This paradox suggests that myostatin inhibition uncouples muscle growth from mitochondrial metabolic optimization, potentially compromising endurance capacity or recovery under high-demand conditions. Importantly, endurance training appears to partially restore mitochondrial lipid composition, suggesting that exercise is a compensatory strategy to preserve mitochondrial quality in the context of myostatin deficiency [54].

## 5. Adipokine Signaling Dynamics in Response to Acute and Chronic Exercise

Adipokines, such as leptin, adiponectin, resistin, chemerin, and pro-inflammatory cytokines (e.g., TNF-α, IL-6), secreted by adipose tissue during physical activity modulate energy metabolism in a context-dependent manner, with sedentary behavior promoting low-grade chronic inflammation, whereas regular exercise reduces adipose tissue mass, promotes its browning, induces an anti-inflammatory profile, and supports metabolic flexibility [75].

A study conducted on 22 obese individuals (BMI 31.7 ± 4.4 kg∙m^−2^) subjected to physical exercise at an 80% VO_2_ peak, which evaluated a panel of 92 inflammation-related cytokines both after the first exercise session and after 8 weeks, revealed that the levels of anti-inflammatory interleukins such as IL-1β, IL-4, IL-13, IL-8, IL-10, and IL-15, as well as IL-1 receptor antagonist (IL-1Ra), colony-stimulating factors, TNFα, and chemokine ligands such as C-C motif chemokine ligand 2 (CCL2) and C-X3-C motif chemokine ligand 1 (CX3CL1) were increased in the blood following an acute bout of exercise. Although improvements in fitness and reductions in adiposity were recorded after 8 weeks of training, the acute cytokine response remained comparable, reflecting a similar per-exercise immune reaction despite physiological adaptations [76].

The secretion of leptin and adiponectin is influenced by the type, duration, and intensity of physical exercise. Serum leptin levels do not change significantly following short bouts of acute intense exercise (less than 60 min) or in sessions with an energy expenditure less than 800 kcal. However, significant reductions in leptin are observed with prolonged exercise (over 60 min) or with high caloric expenditure (over 800 kCal) [77,78].

In contrast, adiponectin increased 30 min after intense short-duration exercise (<60 min), but no significant changes were observed after longer-duration exercise. A meta-analysis of 72 controlled trials reported a significant reduction in serum leptin following chronic exercise interventions (for at least 2 weeks), which was associated with a decrease in body fat percentage [79]. Furthermore, another meta-analysis including 25 studies (262 participants for acute exercise, 377 for chronic exercise) revealed a decrease in leptin in both groups. However, the acute response is influenced by pre-exercise nutrition. The study concluded that at least 180 min/week of moderate-intensity exercise or 120 min/week of high-intensity exercise is required to significantly lower serum leptin levels [80].

Leptin regulates energy balance and the metabolism by activating its receptor isoform b (LepRb), which triggers multiple intracellular signaling pathways [81]. The JAK2/STAT3 pathway is essential for the anorexigenic effect of leptin by stimulating the expression of pro-opiomelanocortin (POMC) and inhibiting NPY [81,82]. Simultaneously, leptin activates the PI3K/Akt pathway, which inhibits the transcription factor Forkhead box protein O1 (FoxO1) and supports appetite suppression [83]. In parallel, leptin regulates glucose homeostasis by inhibiting hepatic gluconeogenesis, an effect dependent on IL-6 and hepatic STAT3 [84]. However, in the long term, leptin induces the expression of suppressor of cytokine signaling 3 (SOCS3), a negative feedback inhibitor that blocks STAT3 activation and contributes to leptin and insulin resistance in the context of obesity [85].

Through binding to its specific receptor (AdipoR1), which is predominantly expressed in skeletal muscle, adiponectin activates an intracellular signaling cascade involving AMPK and the p38 MAPK pathway [86]. The activation of AMPK leads to acute metabolic responses, such as increased glucose uptake via the translocation of GLUT4 to the plasma membrane, through a partially insulin-independent mechanism. In parallel, AMPK also initiates longer-term adaptations, including increased FA oxidation through the upregulation of the mitochondrial enzymes involved in β-oxidation. The p38 MAPK pathway contributes primarily to these chronic effects, supporting the transcriptional activation of the genes involved in mitochondrial biogenesis and metabolic remodeling in response to sustained physical activity [86].

The peripheral effects of the adipokines leptin and adiponectin on skeletal muscle and the liver are depicted in Figure 3.

## 6. Crosstalk Outcomes: Lipid Oxidation, Mitochondrial Biogenesis, and Glucose Uptake

During physical exercise, skeletal muscle and adipose tissue engage in complex and dynamic interorgan crosstalk that is essential for maintaining metabolic homeostasis. This communication is mediated by a wide range of myokines and adipokines [87]. Myokines released from contracting muscle fibers stimulate lipolysis in WAT, increasing the availability of circulating free FA acids as energy substrates for active muscles. Irisin released from muscle promotes the browning of WAT, thereby increasing non-shivering thermogenesis and energy availability during exercise [88]. In parallel, adiponectin secreted by adipose tissue enhances insulin sensitivity by activating AMPK signaling in skeletal muscle, thereby promoting glucose uptake and increasing FA oxidation [88].

A key outcome of muscle–adipose crosstalk is the promotion of metabolic flexibility [89]. Regular physical activity enhances metabolic flexibility by stimulating mitochondrial biogenesis, improving substrate transport, and increasing enzymatic activity in skeletal muscle [90]. In contrast, metabolic inflexibility marked by impaired substrate switching is frequently observed in obesity, insulin resistance, and type 2 diabetes (T2DM) [91].

Importantly, aging is also associated with a progressive decline in metabolic flexibility, due to reduced mitochondrial function, chronic low-grade inflammation, and hormonal alterations [75]. This age-related rigidity in fuel selection contributes to sarcopenia, increased adiposity, and a greater risk of metabolic and cardiovascular disease. Enhancing inter-tissue communication through exercise may thus represent a key strategy to preserve metabolic adaptability and systemic homeostasis during aging [14].

Recent comprehensive reviews, notably by Fang et al. (2022), have synthesized evidence on the roles of specific adipokines (specifically adiponectin and spexin) and myokines (mainly irisin and IL 6) in mediating adipose–muscle crosstalk that regulates energy homeostasis in age-related metabolic disorders. Disruption in these signaling pathways contributes to the pathogenesis of sarcopenic obesity, which Fang et al. describe as a convergence of high fat mass and declining muscle function with aging [92].

Aging and sarcopenia can be prevented or alleviated by modulating the molecular pathways involved in maintaining muscle homeostasis and mitochondrial function. For example, resistance training reduces muscle atrophy by inhibiting complement component 1q (C1q)-induced Wnt signaling as well as the PI3K-Akt-tuberous sclerosis complex (TSC) cascade, which regulates the activity of the target of rapamycin complex 1 (mTORC1) via the Ras homolog enriched in the brain (Rheb). Resistance training in mice attenuates muscle fibrosis and atrophy by suppressing C1q-induced Wnt/β-catenin signaling, specifically through the glycogen synthase kinase-3 beta (GSK)-3β/β-catenin axis [93]. The canonical Wnt pathway is activated when Wnt ligands bind to Frizzled and low-density lipoprotein receptor-related proteins (LRP) receptors on the cell membrane. This interaction, which is enhanced by resistance exercise and inhibited by C1q, initiates downstream signaling [94]. In the absence of Wnt, β-catenin is phosphorylated by a destruction complex consisting of GSK-3β, Axin, adenomatous polyposis coli (APC), and casein kinase 1 alpha (CK1α) and is subsequently degraded. When Wnt is present, this complex is inactivated, allowing β-catenin to accumulate and translocate into the nucleus, where it binds T-cell factor (TCF)/lymphoid enhancer-binding factor (LEF), promoting the transcription of genes involved in muscle regeneration and hypertrophy [94].

Exercise supports muscle–brain communication via the release of myokines such as IL-6, insulin-like growth factor 1 (IGF-1), brain-derived neurotrophic factor (BDNF), cathepsin B (CTSB), irisin, and leukemia inhibitory factor (LIF), which help prevent mild cognitive impairment (MCI) and musculoskeletal aging [95]. Although intense exercise markedly increases muscle blood flow, the high rate of oxygen extraction, particularly at the venous end of capillaries, may lead to localized hypoxia, potentially triggering the secretion of these protective myokines and neurotrophic factors [96].

Figure 4 shows the influence of acute and chronic physical exercise on myokine production in skeletal muscle and adipokine secretion in adipose tissue. This schematic highlights how different exercise modalities and durations modulate the release of key signaling molecules, which subsequently act locally and systemically to regulate inflammation, energy metabolism, and overall metabolic homeostasis.

Leptin released from adipose tissue functions as a metabolic gatekeeper of nutritional status, with low circulating levels commonly observed in patients with sarcopenia and in elderly individuals [97]. By activating receptors located in striated muscle, leptin promotes muscle growth, reduces muscle atrophy, and exerts key metabolic effects, such as enhancing intramuscular glucose uptake and stimulating fatty acid β-oxidation. The anabolic effects of leptin on muscle are also mediated indirectly through the insulin-like growth factor-1 (IGF-1) pathway [98].

The leptin-mediated activation of AMPK leads to the phosphorylation and inhibition of acetyl-CoA carboxylase (ACC), a rate-limiting enzyme in de novo fatty acid biosynthesis. This results in a reduction in the levels of malonyl-CoA, an inhibitor of CPT1, the enzyme responsible for shuttling FAs into mitochondria for β-oxidation [99]. Furthermore, AMPK promotes FA oxidation indirectly by stimulating the translocation of fatty acid translocase/cluster of differentiation 36 (FAT/CD36) to the mitochondrial membrane [100]. Leptin is considered a pro-inflammatory adipokine; however, chronic exercise reduces circulating leptin levels and its associated inflammatory burden, thereby contributing to an overall anti-inflammatory phenotype in physically trained individuals [101]. Nevertheless, in situations of overtraining or energy deficiency, very low leptin levels may become maladaptive, impairing immune and endocrine function.

At the cellular level, in myocytes, leptin exerts a proliferative effect by reducing the expression of myostatin, a negative regulator of muscle growth, and increasing the expression of cyclin D1 and proliferating cell nuclear antigen (PCNA), both of which are involved in cell cycle progression and proliferation. The binding of globular adiponectin (gApN) to AdipoR1 induces Ca^2+^ influx and activates Ca^2+^/calmodulin-dependent protein kinase kinase (CaMKK), which in turn phosphorylates and activates AMPK. AMPK activation is also mediated through the adaptor protein containing the PH domain, PTB domain, and leucine zipper motif-1 (APPL1)/liver kinase B1 (LKB1) pathway, where APPL1 interacts directly with AdipoR1 and promotes the cytoplasmic localization of LKB1, a key upstream kinase that activates AMPK [102].

Through deacetylation, AMPK upregulates the expression of PGC-1α via the SIRT1 signaling pathway which in turn modulates FoxO transcriptional activity by repressing the expression of key atrogenes such as atrogin-1 and muscle RING-finger protein-1 (MuRF1), thereby preventing proteasome-mediated muscle protein degradation while simultaneously enhancing mitochondrial biogenesis, antioxidant defense, and metabolic resilience [103]. Additionally, AMPK upregulates the expression of enzymes involved in reducing oxidative stress, particularly stress resulting from increased FAs [104]. Figure 5 illustrates the crosstalk between skeletal muscle, adipose tissue, the liver, and the cardiovascular system, showing how regular physical activity modulates inflammation, glucose and lipid metabolism, and mitochondrial function through circulating signaling molecules.

The anti-inflammatory effects of adiponectin in skeletal muscle are largely attributed to its ability to induce macrophage polarization toward an anti-inflammatory phenotype, characterized by the secretion of interleukin-10 (IL-10) [105].

## 7. Therapeutic Consequences and Pharmacological Management of Metabolic Flexibility in Aging

The modulation of key signaling pathways, such as AMPK–PGC-1α–SIRT1, PPARs, and NF-κB, has direct implications for the prevention and management of age-related metabolic disorders in physical activity-restricted individuals, and particularly with comorbidity presence such as T2DM, obesity and insulin resistance, sarcopenia, or cardiometabolic syndrome. Through improved mitochondrial efficiency, lipid utilization, glucose uptake, and anti-inflammatory effects, the crosstalk mechanisms described could help preserve metabolic health and physical function in older adults and prevent chronic low-grade inflammation, a hallmark of these diseases. To support the optimization of metabolic flexibility in aging, especially as discussed in the context of muscle–fat crosstalk, myokines, and mitochondrial function, the most effective pharmacological strategies should aim to mimic or enhance exercise-induced signaling pathways such as AMPK–PGC-1α–SIRT1 and reduce chronic inflammation.

In aging, where metabolic flexibility declines due to mitochondrial inefficiency, chronic inflammation, and impaired muscle–fat communication, several pharmacological agents have emerged that mimic or enhance the beneficial effects of physical activity and could support the health of physically impaired individuals. For example, metformin and berberine activate AMPK, promoting fatty acid oxidation, mitochondrial biogenesis, and improved insulin sensitivity [106,107]. Notably, a comprehensive meta-analysis of randomized clinical trials found that berberine significantly improves lipid profiles, reduces triglycerides, total cholesterol, and low-density lipoproteins (LDLs), increases high-density lipoproteins (HDLs), and lowers insulin resistance and fasting glucose in patients with metabolic disorders [108]. Experimental studies further demonstrate that berberine enhances skeletal muscle AMPK phosphorylation, SIRT1 activation, and PGC-1α expression, which are key drivers of mitochondrial function and energy metabolism [108]. In aging rodent models, berberine restored mitochondrial complex activities, balanced redox homeostasis, and reduced neuroinflammation, thus supporting its role in mitigating brain aging [109].

Additionally, PPAR agonists, such as pioglitazone (PPAR-γ) and the investigational GW501516 (PPAR-δ), have been shown in clinical and preclinical studies to enhance lipid metabolism and insulin sensitivity [110,111,112]. Both PPAR-γ and PPAR-δ activation may prove beneficial in maintaining metabolic flexibility during aging, as PPAR-γ, expressed in adipose tissue, enhances insulin sensitivity, while PPAR-δ, predominantly found in skeletal muscle, supports oxidative metabolism and promotes mitochondrial biogenesis [113,114]. SIRT1 activators, including resveratrol, along with nicotinamide adenine dinucleotide (NAD^+^) precursors like nicotinamide riboside (NR) and nicotinamide mononucleotide (NMN), further increase mitochondrial health, energy homeostasis, and stress resistance [115]. Resveratrol indirectly stimulates SIRT1 and thereby increases mitochondrial oxidative capacity, while NR and NMN supplementation elevates NAD^+^ and improves muscle mitochondrial density and insulin sensitivity in older adults. Together, these agents target key energy and stress-responsive pathways [AMPK, PPARs, SIRT1/NAD^+^], highlighting them as promising “exercise mimetics” for supporting healthy aging and metabolic resilience.

Emerging strategies also include myostatin inhibitors, which prevent muscle wasting and improve muscle–fat crosstalk and mitochondrial open reading frame of the 12S rRNA-c (MOTS-c), a mitochondrial-derived peptide that mimics exercise signaling via AMPK activation. Specifically, it acts as a potent metabolic regulator, which is increasingly recognized for its role in mimicking the benefits of exercise in sedentary or metabolically impaired individuals. Functioning predominantly through the activation of AMPK, MOTS-c promotes glucose utilization, enhances fatty acid oxidation, and improves insulin sensitivity, which are critical functions that tend to deteriorate with aging and physical inactivity. In skeletal muscle, MOTS-c also appears to influence nuclear gene expressions linked to mitochondrial biogenesis and metabolic adaptation, thereby supporting mitochondrial health. In aged mice, the exogenous administration of MOTS-c improved physical performance and metabolic profiles, partly through its ability to increase NAD^+^ availability and modulate sirtuin pathways, including SIRT1 and SIRT3, both essential regulators of mitochondrial function and stress resistance [116], whilst recent studies have demonstrated that MOTS-c may also exert protective effects against diet-induced obesity and insulin resistance by altering skeletal muscle gene expression related to metabolism and inflammation [117]. These findings suggest that MOTS-c represents a promising pharmacological strategy to restore metabolic flexibility and delay aging-associated decline, particularly in individuals unable to engage in regular physical activity. Nevertheless, it is important to consider that the current evidence on MOTS-c remains predominantly preclinical [118]. In mouse models, MOTS-c administration has been shown to enhance physical performance across various age groups, including when treatment is initiated in older adulthood [119]. In humans, D’Souza et al. (2020) reported age-related declines in plasma MOTS-c levels, while muscle tissue levels paradoxically increased in healthy males—possibly reflecting shifts in muscle fiber type composition with aging [120]. However, clinical data are still limited. MOTS-c analogs, such as CB4211, are currently undergoing early phase 1 trials, and key aspects including long-term safety, dosing regimens, pharmacokinetics, and therapeutic efficacy in aging or disease-specific populations have yet to be established [121]. GLP-1 receptor agonists exert anti-aging and metabolic benefits by activating AMPK and its downstream effector PGC-1α, promoting mitochondrial biogenesis and substrate utilization, while also modulating PPARs to enhance lipid metabolism and insulin sensitivity in aged individuals [122]. Adjunctive anti-inflammatory agents such as omega-3 fatty acids or low-dose acetylsalicylic acid (AAS) can also help to mitigate the chronic low-grade inflammation commonly associated with aging and metabolic dysfunction, but evidence is still very limited; therefore, a further discussion of these and other potential modulators is beyond the scope of this review [123,124,125].

For elderly individuals with limited mobility or contraindications to structured exercise, these pharmacological approaches may serve as “exercise mimetics,” offering a means to preserve metabolic health and prevent frailty-related complications. By targeting key signaling pathways involved in glucose and lipid metabolism, mitochondrial function, and muscle preservation, such interventions could reduce the burden of sarcopenia, insulin resistance, T2DM, and cardiometabolic syndrome. Incorporating these agents into personalized geriatric care strategies, particularly when supported by biomarker-guided assessments, has the potential to delay or prevent the onset of metabolic and neurodegenerative diseases, while improving the quality of life in older populations unable to maintain regular physical activity. The pharmacological modulation of convergent signaling pathways involved in metabolic flexibility is summarized in Figure 6.

Together, these agents converge on shared molecular nodes such as AMPK, PGC-1α, SIRT1, and PPARs, thus creating a network that supports metabolic adaptability, mitochondrial quality control, and reduced inflammatory load. This integrated approach may be particularly valuable for elderly individuals who are unable to perform regular physical activity, offering a pharmacological strategy to mimic the systemic benefits of exercise and mitigate the risk of age-related metabolic and neurodegenerative diseases.

In addition to their role as exercise mimetics, several of these pharmacological agents may also enhance exercise performance, thereby amplifying exercise-mediated benefits in older adults. For example, AMPK activators such as metformin and berberine improve mitochondrial efficiency and energy substrate utilization, which can reduce fatigue and increase endurance capacity [106,107]. Similarly, SIRT1 activators like resveratrol and NAD^+^ precursors have been shown to enhance mitochondrial biogenesis and oxidative metabolism, leading to improved muscle function and recovery [112,118,126]. Mitochondrial-derived peptides such as MOTS-c may further promote fatigue resistance and enhance muscle strength by activating key metabolic pathways and supporting mitochondrial health [117]. These adaptations could help older adults tolerate and sustain regular physical activity, thus maximizing the combined benefits of exercise and pharmacological support. For example, the best pharmacological approach to preserve metabolic flexibility in aging T2DM patients who cannot exercise involves combining metformin or berberine (AMPK activation), NAD^+^ precursors or resveratrol (mitochondrial support), and anti-inflammatory agents (omega-3s or low-dose AAS). This integrative strategy targets energy metabolism, inflammation, and mitochondrial resilience, mimicking some of the beneficial effects of physical activity.

For stratifying exercise ‘mimicry’ dependent on individual health and comorbidity, pharmacological intervention would need to be tailored to the patient as shown in Table 1.

This table presents a stratified pharmacological algorithm tailored for aging individuals with T2DM who have limited ability to engage in physical activity. The rationale centers on the pharmacological mimicry of exercise-induced metabolic benefits. Agents are chosen based on their ability to modulate key molecular pathways—AMPK, PPARs, SIRT1/PGC-1α, and NF-κB—linked to energy balance, mitochondrial function, inflammation, and glucose homeostasis. Personalized therapy recommendations are made by aligning agent mechanisms to common comorbidities such as cardiovascular disease, chronic inflammation, renal impairment, and frailty. This strategy enhances metabolic flexibility, offering a framework for targeted intervention in geriatric T2DM management.

## 8. Conclusions

This narrative review highlights the critical interactions between aging, metabolic flexibility, and the potential of pharmacological mimetics to simulate exercise-induced benefits, particularly in older adults and individuals with T2DM who are unable to engage in regular physical activity. A central theme is the modulation of key signaling nodes—AMPK, PPARs, and the SIRT1–PGC-1α axis—by compounds such as metformin, berberine, pioglitazone, resveratrol, and NAD^+^ precursors. The review suggests a possible therapeutic role of these agents in promoting fatty acid oxidation, improving mitochondrial function, and mitigating chronic low-grade inflammation. Importantly, the inclusion of emerging molecules such as MOTS-c, a mitochondrial-derived peptide with potent AMPK-activating capacity, represents a novel direction in the pharmacological mimicry of exercise. This integrated view advances our understanding of how tailored drug regimens can support energy homeostasis, muscle–fat crosstalk, and cellular resilience during aging.

Another novel aspect of this review lies in the stratification of treatment options according to specific comorbidities common in older adults, such as cardiovascular disease, sarcopenia, and cognitive decline. By mapping distinct pharmacodynamic profiles to patient-specific needs, the review proposes a precision-based framework for protecting metabolic plasticity. Additionally, it reinforces the concept that multifaceted interventions targeting inflammation (e.g., omega-3 FAs, low-dose AAS) and mitochondrial biogenesis (e.g., PGC-1α inducers) may offer synergistic benefits. While these findings are promising, it is important to acknowledge the current limitations, including individual variability and the unproven long-term safety of exercise mimetics. This work lays the groundwork for future clinical translation and the development of combination therapies tailored to preserve metabolic health in vulnerable populations.

## 9. Limitations and Future Research Directions

Although increased attention has recently been given to studies focused on maintaining metabolic flexibility through physical activity in aging, and significant progress has been made in elucidating the molecular mediators of muscle–fat crosstalk, several limitations remain. Much of the current evidence is based on preclinical models or short-term clinical studies, which limits its extrapolation to older adults with multiple comorbidities. Furthermore, the long-term safety and efficacy of pharmacological “exercise mimetics”, such as AMPK activators, PPAR agonists, myostatin inhibitors, and mitochondrial peptides, require further validation in diverse, aging populations.

Moreover, interindividual variability in response to both exercise and pharmacological interventions is poorly understood and may be influenced by genetic background, microbiome composition, sex differences, and epigenetic factors. The translational gap between molecular findings and clinically applicable therapies remains substantial.

Future research should prioritize large-scale, longitudinal studies that incorporate biomarker-guided stratification, multi-omics approaches, and precision combinations of exercise and pharmacotherapy. Special attention should be given to evaluating interventions that concurrently target inflammation, mitochondrial function, and hormonal regulation. Additionally, clinical trials specifically designed for frail, sedentary, or mobility-impaired elderly individuals are needed to assess the therapeutic applicability of the proposed strategies.

## Figures and Tables

**Figure 1 pharmaceuticals-18-01222-f001:**
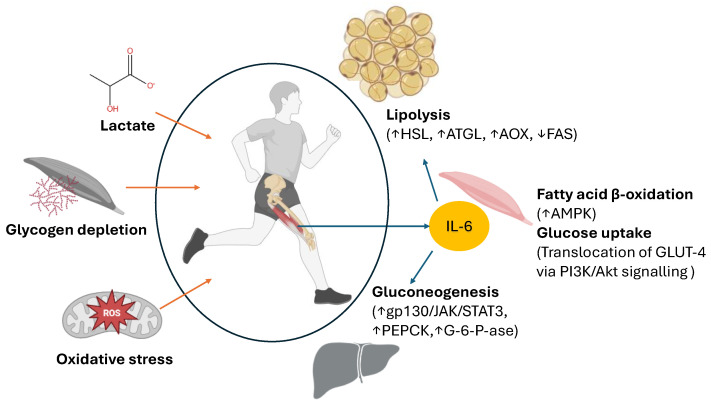
Metabolic effect of the myokine IL-6 released during exercise. Muscle activity leads to lactic acid formation in skeletal muscle because of glycolysis, along with the depletion of intramuscular glycogen stores. Owing to the intensification of mitochondrial oxidative metabolism, there is an increased production of reactive oxygen species (ROS) and oxidative stress. The red arrows indicate the main promoters of IL-6 release from the muscle during exercise. The release of IL-6 into the circulation promotes the mobilization of fatty acids from adipose tissue by activating enzymes such as hormone-sensitive lipase (HSL), adipose triglyceride lipase (ATGL), and α-ketoglutarate dehydrogenase (AOX), while simultaneously inhibiting fatty acid synthesis via the inhibition of fatty acid synthase (FAS). In the liver, IL-6 stimulates gluconeogenesis through activation of the gp130/JAK/STAT3 signaling pathway and activation of key gluconeogenic enzymes, including phosphoenolpyruvate carboxykinase (PEPCK) and glucose-6-phosphatase (G-6-P-ase). In skeletal muscle, IL-6-induced activation of AMPK facilitates GLUT-4 translocation and glucose uptake into muscle cells through an insulin-independent mechanism, while also promoting β-oxidation of fatty acids, particularly short-chain fatty acids, for aerobic ATP production. ↑ up arrows mean increase and ↓ down arrows mean decrease. This image was created using BioRender (https://BioRender.com/n4xejkf, accessed on 18 August 2025).

**Figure 2 pharmaceuticals-18-01222-f002:**
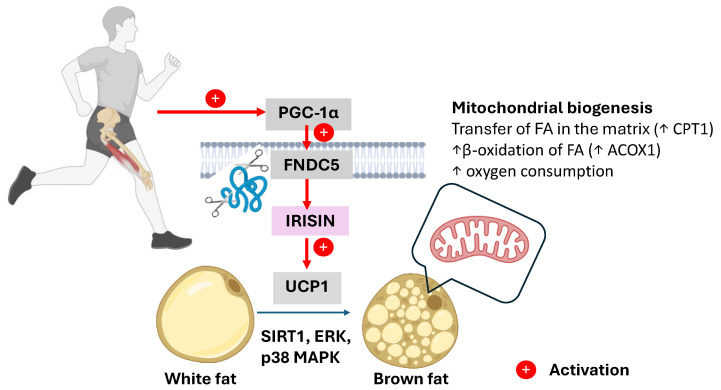
Exercise-induced irisin release promotes browning of white adipose tissue. Physical activity activates PGC-1α expression in skeletal muscle, leading to the cleavage of FNDC5 into the circulating myokine irisin. Irisin acts on white adipocytes to upregulate UCP1, a key thermogenic marker, and initiates mitochondrial biogenesis. This signaling cascade results in the browning of WAT, characterized by increased mitochondrial concentration and enhanced energy dissipation as heat. (PGC-1α—peroxisome proliferator-activated receptor gamma coactivator-1 alpha, FNDC5—fibronectin type III domain-containing protein 5, UCP1—uncoupling protein 1, CPT1—carnitine palmitoyltransferase 1, FAs—fatty acids, ACOX1—acyl-CoA oxidase 1, SIRT1—sirtuin 1, p38 MAPK—p38 mitogen-activated protein kinase, ERK—extracellular signal-regulated kinase, ↑ up arrows mean increase). This image was created using BioRender. (https://BioRender.com/67zf659, accessed on 18 August 2025).

**Figure 3 pharmaceuticals-18-01222-f003:**
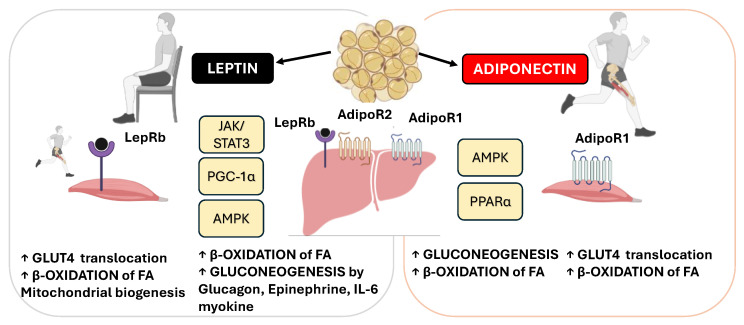
Leptin and adiponectin signaling pathways in skeletal muscle and liver. This schematic illustrates the peripheral actions of the adipokines leptin and adiponectin on skeletal muscle and the liver in the context of physical activity and metabolic regulation. Leptin, which is secreted by adipose tissue, binds to its long-form LepRb in skeletal muscle and the liver. In skeletal muscle, leptin activates the JAK/STAT3, PGC-1α, and AMPK signaling pathways, promoting mitochondrial biogenesis, glucose uptake, and fatty acid oxidation. Leptin, through binding to hepatic LepRb receptors, can inhibit gluconeogenesis and improve lipid metabolism under resting and postprandial conditions. However, during acute physical exercise, this effect is counterbalanced by stress hormones and myokines such as IL-6, which stimulate gluconeogenesis to ensure the necessary ATP supply for active skeletal muscle. Adiponectin, which acts primarily through its receptor AdipoR1 in muscle, stimulates AMPK and PPARα signaling, enhancing glucose uptake, fatty acid oxidation, and mitochondrial function. (LepRb—leptin receptor isoform b, JAK—Janus kinase, STAT3—signal transducer and activator of transcription 3, PGC-1α—peroxisome proliferator-activated receptor gamma coactivator 1-alpha, AMPK—AMP-activated protein kinase, IL-6—interleukin-6, ATPadenosine triphosphate, AdipoR1—adiponectin receptor 1, PPARα—peroxisome proliferator-activated receptor alpha, FAs—fatty acids, ↑ up arrows mean increase). This image was created using BioRender (https://BioRender.com/d3aik7f, accessed on 18 August 2025).

**Figure 4 pharmaceuticals-18-01222-f004:**
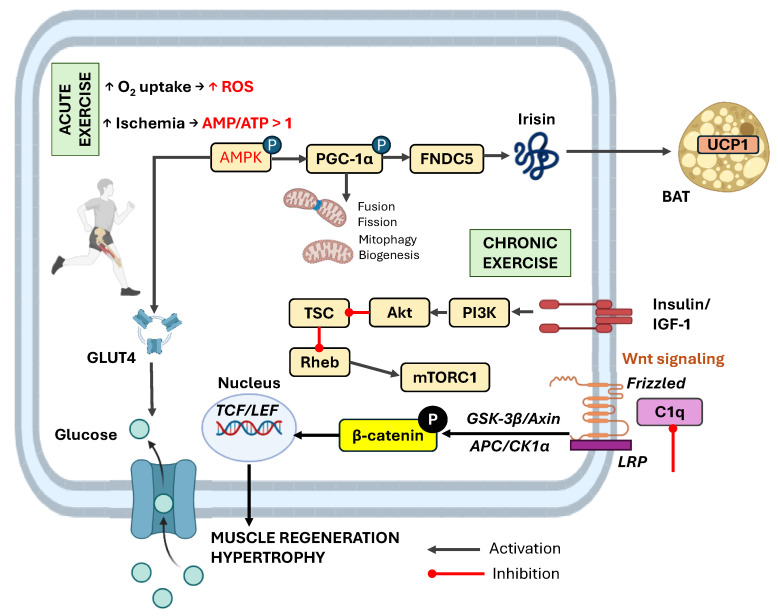
Molecular mechanisms through which exercise regulates skeletal muscle metabolism, mitochondrial quality, and adipose tissue remodeling. Acute exercise increases oxygen consumption and hypoxic stress, leading to elevated ROS and an increased AMP/ATP ratio, which activates AMPK. Activated AMPK phosphorylates and upregulates PGC-1α, promoting mitochondrial quality control via biogenesis, mitophagy, fusion, and fission. PGC-1α also induces FNDC5 expression, resulting in the secretion of irisin, a myokine that stimulates UCP1 expression and browning in white adipose tissue. In parallel, exercise promotes glucose uptake through GLUT4 translocation to the plasma membrane. Chronic exercise activates the PI3K–Akt–mTORC1 pathway by inhibiting the TSC and activating Rheb, thereby supporting protein synthesis and muscle hypertrophy. Resistance training also suppresses C1q-induced Wnt signaling, which otherwise contributes to muscle atrophy and fibrosis. In the absence of Wnt, β-catenin is degraded via a complex that includes GSK-3β, Axin, APC, and CK1α. Wnt activation inactivates this complex, allowing β-catenin to enter the nucleus, bind TCF/LEF, and drive the transcription of genes involved in muscle growth and repair. Together, these pathways contribute to increased metabolic flexibility, improved mitochondrial function, and the prevention of sarcopenia. (AMPK—AMP-activated protein kinase, PGC-1α—Peroxisome proliferator-activated receptor gamma coactivator 1-alpha, FNDC5—Fibronectin type III domain-containing protein 5, UCP1—Uncoupling protein 1, GLUT4—Glucose transporter type 4, ROS—Reactive oxygen species, PI3K—Phosphoinositide 3-kinase, Akt—Protein kinase B, TSC—Tuberous sclerosis complex, Rheb—Ras homolog enriched in brain, mTORC1—Mechanistic target of rapamycin complex 1, BAT—Brown adipose tissue, C1q—Complement component 1q, GSK-3β—Glycogen Synthase Kinase-3 Beta, APC—Adenomatous Polyposis Coli, CK1α—Casein Kinase 1 alpha, TCF—T-cell Factor, LEF—Lymphoid Enhancer-binding Factor, ↑ up arrows mean increase). This image was created using BioRender (https://biorender.com/yhzyxwv, accessed on 18 August 2025).

**Figure 5 pharmaceuticals-18-01222-f005:**
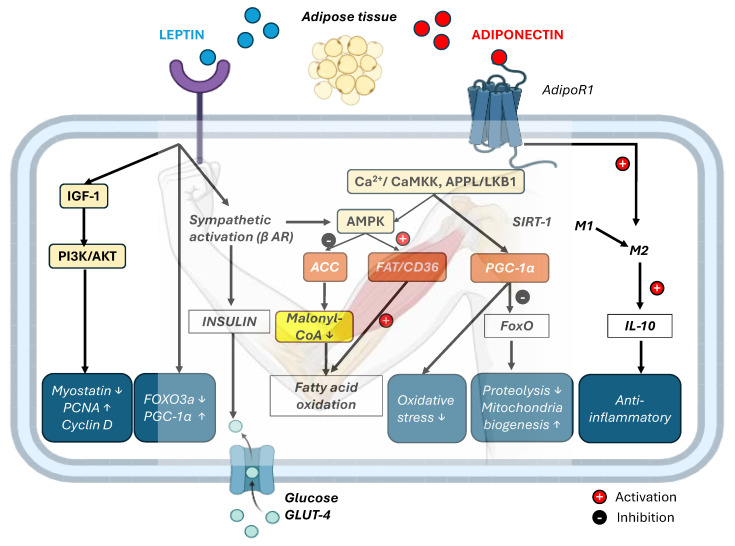
The coordinated effects of leptin and adiponectin on skeletal muscle metabolism, mitochondrial function, and inflammation. This figure illustrates the regulation of skeletal muscle physiology by leptin and adiponectin. Leptin exerts anabolic and metabolic effects through the activation of the PI3K/Akt pathway via IGF-1, resulting in the suppression of myostatin, increased expression of PCNA and cyclin D, and downregulation of FoxO3a, which reduces atrophy-related gene expression. Leptin also enhances PGC-1α expression and increases glucose uptake by promoting insulin sensitivity and GLUT4 translocation. Additionally, leptin stimulates sympathetic activation (via β-AR), which leads to AMPK activation and downstream inhibition of ACC, reducing malonyl-CoA levels and facilitating fatty acid oxidation. Adiponectin binds to AdipoR1, triggering Ca^2+^ influx and activation of the CaMKK–APPL–LKB1 pathway, which also activates AMPK. This in turn enhances FA transport (FAT/CD36), improves the oxidative stress response, and upregulates PGC-1α, promoting mitochondrial biogenesis and inhibiting FoxO-mediated proteolysis. Through SIRT1, adiponectin further modulates FoxO activity, suppressing proteolytic gene expression. In parallel, adiponectin promotes macrophage polarization toward the anti-inflammatory M2 phenotype, increasing IL-10 secretion and contributing to a protective, anti-inflammatory muscle microenvironment. (IGF-1—Insulin-like growth factor-1, PI3K—Phosphoinositide 3-kinase, Akt—Protein kinase B, PCNA—Proliferating cell nuclear antigen, FoxO—Forkhead box O, AMPK—AMP-activated protein kinase, ACC—Acetyl-CoA carboxylase, FAT/CD36—Fatty acid translocase/Cluster of differentiation 36, PGC-1α—Peroxisome proliferator-activated receptor gamma coactivator-1 alpha, SIRT1—Sirtuin 1, APPL—Adaptor protein containing PH domain, PTB domain and leucine zipper motif, LKB1—Liver kinase B1, CaMKK—Ca^2+^/calmodulin-dependent protein kinase kinase, IL-10—Interleukin-10, M1/M2—Pro-inflammatory (M1) and anti-inflammatory (M2) macrophage phenotypes, β AR—Beta-adrenergic receptor, ↑ up arrows mean increase and ↓ down arrows mean decrease). This image was created using BioRender (https://BioRender.com/r0klswn, accessed on August 2025).

**Figure 6 pharmaceuticals-18-01222-f006:**
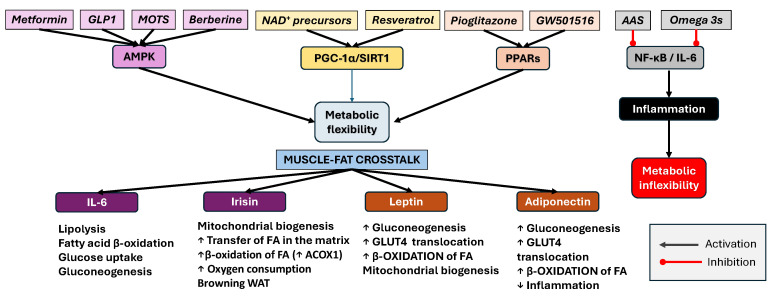
Pharmacological targeting of convergent signaling pathways to enhance metabolic flexibility in aging: an integrated mechanistic model. This figure presents a conceptual framework summarizing how diverse pharmacological agents modulate key molecular pathways to preserve metabolic flexibility, particularly in older individuals with limited physical activity. Metabolic flexibility, the capacity to efficiently switch between glucose and lipid utilization in response to energy demands, is governed by the dynamic interplay of mitochondrial function, nutrient sensing, muscle–fat crosstalk, and low-grade inflammation. The diagram shows how different drug classes exert complementary and sometimes overlapping effects on these regulatory nodes. Metformin and berberine activate AMPK, a central metabolic sensor that enhances glucose uptake, FA oxidation, and mitochondrial biogenesis. PPAR agonists, such as pioglitazone (PPAR-γ) and GW501516 (PPAR-δ, experimental), further promote insulin sensitivity, lipid transport, and oxidative metabolism by upregulating genes involved in mitochondrial function. The SIRT1/PGC-1α axis, activated by compounds such as resveratrol and NAD^+^ precursors (e.g., NR, NMN), promotes mitochondrial health, oxidative phosphorylation, and cellular resilience to stress. Emerging agents that mimic exercise-induced signaling include myostatin inhibitors, which reduce muscle catabolism and enhance myokine secretion (e.g., irisin) leading to white adipose tissue browning and improved thermogenic capacity. Similarly, MOTS-c, a mitochondrial-derived peptide, functions as a potent AMPK activator, improving glucose metabolism and lipid oxidation. In parallel, anti-inflammatory agents such as omega-3 polyunsaturated FAs and low-dose aspirin mitigate chronic low-grade inflammation by attenuating NF-κB and IL-6 signaling, indirectly supporting insulin action and mitochondrial function (AMPK—AMP-activated protein kinase, IL-6—interleukin-6, NF-κB—nuclear factor-κB, MOTS-c—mitochondrial open reading frame of the 12S rRNA type-c, FAs—fatty acids, SIRT1—sirtuin, PGC-1α—peroxisome proliferator-activated receptor gamma coactivator 1-alpha, NAD^+^—nicotinamide adenine dinucleotide, NR—nicotinamide riboside, NMN—nicotinamide mononucleotide, PPAR—peroxisome proliferator-activated receptor, ↑ up arrows mean increase and ↓ down arrows mean decrease).

**Table 1 pharmaceuticals-18-01222-t001:** Recommended pharmacological agents and exercise strategies for aging individuals, stratified by diagnosed clinical conditions (T2DM, cardiovascular disease) and relevant functional or metabolic traits (frailty, chronic inflammation, renal impairment) that influence metabolic flexibility, exercise tolerance, or treatment safety in older adults.

Clinical Condition or Functional/Metabolic Trait	Recommended Agents	Mechanism/Target	Recommended Exercise Protocol	Notes	Ref.
Cardiovascular disease	Metformin, Omega 3s, Low-dose AAS	AMPK activation, anti-inflammatory, antiplatelet	Moderate-intensity aerobic exercise (e.g., 30–40 min, 4–5 days/week)	Bleeding risk of AAS	[127,128,129,130]
Chronic inflammation (↑CRP/IL-6)	Berberine, Omega 3s	AMPK activation, NF-κB inhibition	Combination of resistance (2–3 days/week) and aerobic exercise (3–4 days/week)	CRP-guided personalization	[131,132,133]
Mitochondrial disfunction/Sarcopenia	NAD^+^ precursors (NR/NMN), Resveratrol	SIRT1-PGC-1α axis, Oxidative phosphorilation	Progressive resistance training (2–3 days/week) with balance/flexibility exercises	Exercise mimetics (like MOTS) to be considered	[118,134,135,136]
Poor glycemic control	Metformin, Pioglitazone, GLP1 analogs	AMPK + PPARγ insulin sensitization	HIIT or moderate-intensity continuous training (≥150 min/week)	Fluid retention and edema with pioglitazone	[137,138]
Renal impairment (* eGFR < 30)	Resveratrol, Omega 3s	Antioxidant, mitochondrial protection	Low-impact aerobic training (walking, cycling) 20–30 min/day	Avoid metformin and pioglitazone	[139,140,141]
Frailty	NAD^+^ precursors, Resveratrol, MOTS	Mitochondrial protection, muscle maintenance	Low-intensity combined training (resistance, balance, and aerobic exercise 2–3 days/week)	Experimental options, in early human trials	[142,143,144,145]

* eGFR—estimated glomerular filtration rate, ↑ up arrows mean increase.

## Data Availability

No new data were created or analyzed in this study. Data sharing is not applicable to this article.

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
