# Peer review of "Exercise-Induced Muscle–Fat Crosstalk: Molecular Mediators and Their Pharmacological Modulation for the Maintenance of Metabolic Flexibility in Aging"

_pharmaceuticals, 2025, doi:10.3390/ph18081222_

Round 1
Reviewer 1 Report
Comments and Suggestions for Authors
Tero-Vescan et al., discussed the role of exercise-induced myokines that help cross-talk of muscle and fat to target various age-related diseases. However, several papers have already been published on the same topic (PMID: 30319436; PMID: 32393961). Additionally, the authors casually wrote this review manuscript without providing proper references throughout the manuscript. This issue must be addressed throughout the whole manuscript. I've included my detailed comments below.
- Add the ref for line no 54.
- This cycle also leads to an increase in the production 59 of reactive oxygen species (ROS), which, while capable of damaging cellular components, 60 are essential for mitophagy and apoptosis in senescent cells [ What are all the cellular components- authors should mention for better clarification]
- Include the ref for line no 92
- It is confusing how glycogen and lipids consume ATP? Does the author mean the breakdown of glycogen to ATP synthesis? (Simultaneously, it inhibits ATP-consuming processes, including glycogen, lipid, and protein synthesis, as well as cell growth and proliferation)
- In signalling terms, (What does the author mean by “ signaling term”?)
- Support with a suitable reference for the following text “strongly influenced by training status”
- Support the text with proper references for line no:136-138
- sonly a marker of mitochondrial dysfunction? What does the author intend to say here?
- No supporting evidence is given in the manuscript for “Experimental evidence from PGC-1α knockout mice revealed a sharp decline in mitochondrial content and exercise performance”
- Again, no reference for “Multiomics analyses and human cohort studies further show that trained individuals”
- Although the figures are nicely represented about he concepts of exercise-induced myokines, no figures are cited within the text. Why were the authors casually prepared the manuscript? Also, include the names of the signaling molecules inside the figures in the caption.
- Again, line no. 226,228 studies have shown that……. the authors included only one study?
- ther studies have shown that intravenous administration of IL-6 at 234 higher concentrations (>100 pg/mL, comparable to those used during strenuous exercise) 235 stimulates lipolysis….. No references…
- in the context of T2DM. NO reference
- involving 77 4- to 12-year-old obese prepubertal children… What is this? 774 is the age of the children?
- The modulation of these pathways? What are these pathways? How can readers understand which pathway the authors intend to say?
- Add reference for “For example, metformin and berberine activate AMPK, promoting fatty acid oxidation, mitochon- 603 drial biogenesis, and improved insulin sensitivity”
- Add reference for “have shown in clinical and preclinical studies to enhance lipid metabolism and insulin sensitivity”
- In conclusion, the authors mention that pharmacological mimetics can stimulate exercise-induced benefits in various diseases. However, in the texts, authors did not discuss how these agents can improve exercise performance to increase the exercise-mediated benefits in the aging group.
- Additionally, in Table 1, the authors did not provide an exercise protocol along with pharmacological agents that target or improve signaling molecules such as AMPK, PGC-1α, and Sir1. Authors should include the use of these agents with exercise protocols in the table to mimic these signaling molecules in different age-related diseases
Author Response
Dear Respected Reviewer 1,
We would like to sincerely thank you for your careful and thoughtful evaluation of our manuscript. We greatly appreciate the time and effort dedicated to providing constructive feedback, which has helped improve the quality and clarity of the article.
Kindly find the revisions made to the manuscript in accordance with your suggestions, highlighted using the track changes feature.
Tero-Vescan et al., discussed the role of exercise-induced myokines that help cross-talk of muscle and fat to target various age-related diseases. However, several papers have already been published on the same topic (PMID: 30319436; PMID: 32393961). Additionally, the authors casually wrote this review manuscript without providing proper references throughout the manuscript. This issue must be addressed throughout the whole manuscript. I've included my detailed comments below.
- Add the ref for line no 54.
We thank the reviewer for this observation. A reference has now been added to support the statement on line 54. The relevant citation has been inserted in the revised manuscript and included in the reference list.
Reuben L Smith, et al. Metabolic Flexibility as an Adaptation to Energy Resources and Requirements in Health and Disease, Endocrine Reviews, Volume 39, Issue 4, August 2018, Pages 489–517, https://doi.org/10.1210/er.2017-00211
- This cycle also leads to an increase in the production of reactive oxygen species (ROS), which, while capable of damaging cellular components, are essential for mitophagy and apoptosis in senescent cells. What are all the cellular components- authors should mention for better clarification.
We thank the reviewer for this helpful suggestion. In the revised manuscript, we have clarified which cellular components are affected by ROS, specifying membrane lipids, proteins, and DNA, to enhance the scientific accuracy and clarity of the statement. The paragraph in the revised version is:
“This cycle also leads to an increase in the production of reactive oxygen species (ROS), which, while capable of damaging cellular components, including membrane lipids, structural and enzymatic proteins, and both nuclear and mitochondrial DNA, also play essential roles in signalling processes such as mitophagy and apoptosis in senescent cells. During the anabolic phase, the activation of mechanistic target of rapamycin (mTOR)-dependent pathways supports protein synthesis and muscle remodelling.”
- Include the ref for line no 92
We thank the reviewer for highlighting the need for a supporting citation. The following citation has been added to support this assertion:
Egan B, Sharples AP. Molecular responses to acute exercise and their relevance for adaptations in skeletal muscle to exercise training. Physiol Rev. 2023 Jul 1;103(3):2057-2170. doi: 10.1152/physrev.00054.2021. Epub 2022 Nov 17. PMID: 36395350.
- It is confusing how glycogen and lipids consume ATP? Does the author mean the breakdown of glycogen to ATP synthesis? (Simultaneously, it inhibits ATP-consuming processes, including glycogen, lipid, and protein synthesis, as well as cell growth and proliferation)
We thank the reviewer for pointing out this ambiguity. We agree that the original phrasing may be unclear. To clarify, we were referring to the synthesis (not breakdown) of glycogen and lipids, which are anabolic, ATP-consuming processes. The sentence has been revised accordingly to avoid confusion:
“AMPK activation, triggered by ATP/AMP < 1 ratio, stimulates ATP-generating processes such as fatty acid oxidation, glycolysis and glucose transport in muscle cells [6]. Simultaneously, it inhibits ATP-consuming processes including glycogen, lipid, and protein synthesis, as well as cell growth and proliferation [7].”
- In signalling terms, (What does the author mean by “ signaling term”?)
We thank the reviewer for this observation. We agree that the phrase “in signalling terms” may be vague or unnecessarily informal. To improve clarity, we have revised the sentence to specify that the focus is on molecular signalling pathways activated during and after exercise, particularly those regulating mitochondrial biogenesis and quality control processes such as autophagy and mitophagy. This is the revised phrase:
“In regards to molecular signalling, PGC-1α activation during physical exercise induces mitochondrial biogenesis, whereas the postexercise recovery phase triggers autophagy and mitophagy to eliminate senescent, dysfunctional, or ROS-damaged mitochondria [10].”
- Support with a suitable reference for the following text “strongly influenced by training status”
We thank the reviewer for this insightful request. A recent publication has been added to support the assertion that trained individuals exhibit markedly enhanced mitochondrial adaptations to exercise compared to untrained peers, regardless of age.
Mesquita PHC, et al. Skeletal Muscle Ribosome and Mitochondrial Biogenesis in Response to Different Exercise Training Modalities. Front Physiol. 2021 Sep 10;12:725866. doi: 10.3389/fphys.2021.725866.
- Support the text with proper references for line no:136-138
We thank the reviewer for this observation. We have added appropriate recent references to support the statement regarding the mechanisms of ROS production during exercise (lines 136–138). These references highlight both the classical concept of mitochondrial inefficiency and the emerging understanding of alternative sources and regulatory processes. The content in lines 136-138 was rephrased as:
"Various mechanisms have been proposed to explain ROS production during exercise, which, although not yet fully understood, has traditionally been attributed to inefficient mitochondrial respiration; however, emerging evidence suggests that the underlying processes are far more complex."
The following references were inserted:
- Powers SK, Jackson MJ. Exercise-induced oxidative stress: cellular mechanisms and impact on muscle force production. Physiol Rev. 2008 Oct;88(4):1243-76. doi: 10.1152/physrev.00031.2007. PMID: 18923182; PMCID: PMC2909187.
- Sies, H., Jones, D.P. Reactive oxygen species (ROS) as pleiotropic physiological signalling agents. Nat Rev Mol Cell Biol 21, 363–383 (2020). https://doi.org/10.1038/s41580-020-0230-3
- sonly a marker of mitochondrial dysfunction? What does the author intend to say here?
We thank the reviewer for pointing this out. The word “sonly” was a typographical error. The correct wording is “only a marker of mitochondrial dysfunction”, indicating that the biomarker in question is not used exclusively as an indicator of mitochondrial dysfunction but also plays a regulatory role in cellular adaptation. We have corrected this typo in the revised manuscript for clarity.
This paradox explains why exercising muscle generates more ROS despite more efficient coupling and supports the idea that ROS production is not only a marker of mitochondrial dysfunction, but also a regulated physiological signal, particularly involved in adaptation via redox-sensitive pathways such as PGC-1α, AMPK, and nuclear factor kappa-light-chain-enhancer of activated B cells (NF-κB) [19]
- No supporting evidence is given in the manuscript for “Experimental evidence from PGC-1α knockout mice revealed a sharp decline in mitochondrial content and exercise performance”
We thank the reviewer for this important observation. We have now added appropriate supporting references to substantiate the statement regarding the decline in mitochondrial content and exercise performance observed in PGC-1α knockout (KO) mice.
- Vainshtein A, Tryon LD, Pauly M, Hood DA. Role of PGC-1α during acute exercise-induced autophagy and mitophagy in skeletal muscle. Am J Physiol Cell Physiol. 2015 May 1;308(9):C710-9. doi: 10.1152/ajpcell.00380.2014. Epub 2015 Feb 11. PMID: 25673772; PMCID: PMC4420796.
- Geng T, Li P, Okutsu M, Yin X, Kwek J, Zhang M, Yan Z. PGC-1alpha plays a functional role in exercise-induced mitochondrial biogenesis and angiogenesis but not fiber-type transformation in mouse skeletal muscle. Am J Physiol Cell Physiol. 2010 Mar;298(3):C572-9. doi: 10.1152/ajpcell.00481.2009. Epub 2009 Dec 23. PMID: 20032509; PMCID: PMC3353735.
- Again, no reference for “Multiomics analyses and human cohort studies further show that trained individuals”
We thank the reviewer for this valuable suggestion. In the revised manuscript, we have added a specific reference to support this assertion.
Jacques M, et al. Molecular landscape of sex- and modality-specific exercise adaptation in human skeletal muscle through large-scale multi-omics integration. Cell Rep. 2025 Jun 24;44(6):115750. doi: 10.1016/j.celrep.2025.115750. Epub 2025 May 28. PMID: 40445834.
- Although the figures are nicely represented about he concepts of exercise-induced myokines, no figures are cited within the text. Why were the authors casually prepared the manuscript? Also, include the names of the signaling molecules inside the figures in the caption.
We thank the reviewer for this helpful comment. In the revised manuscript, we have now ensured that each figure is explicitly cited in the appropriate sections.
Regarding introducing the names of the signalling molecules inside the figures in the caption, we would like to clarify that the names and full forms of all signalling molecules depicted in the figures are already provided at the end of each figure’s explanatory text. Adding the same details to the caption itself would make the captions excessively long and repetitive. For clarity and readability, we have maintained the abbreviations in the figures and provided the complete names and definitions in the explanatory text below each figure. However, if the reviewer considers this addition essential, we will be glad to include the full names of all signalling molecules directly in the captions.
- Again, line no. 226,228 studies have shown that……. the authors included only one study?
Thank you for your observation. Additional relevant references have now been added to support the statement on lines 226 and 228.
- Nash D, et al. IL-6 signaling in acute exercise and chronic training: Potential consequences for health and athletic performance. Scand J Med Sci Sports. 2023 Jan;33(1):4-19. doi: 10.1111/sms.14241. Epub 2022 Oct 8. PMID: 36168944; PMCID: PMC10092579.
- Ostrowski K, Schjerling P, Pedersen BK. Physical activity and plasma interleukin-6 in humans--effect of intensity of exercise. Eur J Appl Physiol. 2000 Dec;83(6):512-5. doi: 10.1007/s004210000312. PMID: 11192058.
- ther studies have shown that intravenous administration of IL-6 at 234 higher concentrations (>100 pg/mL, comparable to those used during strenuous exercise) 235 stimulates lipolysis….. No references…
Thank you for your comment. We have revised the text accordingly: the word 'studies' has been changed to 'study' because the data refer to a single study by Wedell-Neergaard et al (2018).
Wedell-Neergaard AS, et al. Exercise-Induced Changes in Visceral Adipose Tissue Mass Are Regulated by IL-6 Signaling: A Randomized Controlled Trial. Cell Metab. 2019 Apr 2;29(4):844-855.e3. doi: 10.1016/j.cmet.2018.12.007. Epub 2018 Dec 27. PMID: 30595477.
- in the context of T2DM. NO reference
Thank you for your comment. A relevant reference has now been added to support the statement regarding the phrase: The effects of IL-6 on glucose metabolism are largely attributed to exercise-mediated activation of AMPK, an effect that is not observed for IL-6 released in the context of T2DM.
- Jiang, L.Q.; Duque-Guimaraes, D.E.; Machado, U.F.; Zierath, J.R.; Krook, A. Altered Response of Skeletal Muscle to IL-6 in Type 2 Diabetic Patients. Diabetes 2013, 62, 355–361, doi:10.2337/db11-1790.
- involving 77 4- to 12-year-old obese prepubertal children… What is this? 774 is the age of the children?
Thank you for your comment. We have rephrased the sentence to avoid any ambiguity regarding the number and age of the children. The corrected sentence now reads as follows:
"A study by Herouvi et al. (2025), involving 77 obese prepubertal children aged 4 to 12 years, reported elevated serum irisin levels and potential irisin resistance in obese participants compared with healthy controls."
- The modulation of these pathways? What are these pathways? How can readers understand which pathway the authors intend to say?
Thank you for your valuable comment. We have clarified the sentence by explicitly naming the pathways to avoid ambiguity. The revised text now reads:
"The modulation of key signalling pathways, such as AMPK–PGC-1α–SIRT1, PPARs, and NF-κB, has direct implications for the prevention and management of age-related metabolic disorders, in physical activity restricted individuals, and particularly with comorbidity presence such as T2DM, obesity and insulin resistance, sarcopenia or cardiometabolic syndrome."
- Add reference for “For example, metformin and berberine activate AMPK, promoting fatty acid oxidation, mitochon- 603 drial biogenesis, and improved insulin sensitivity”
Thank you for your comment. We have now added a reference to support this statement.
- Wu S, Zou MH. AMPK, Mitochondrial Function, and Cardiovascular Disease. Int J Mol Sci. 2020 Jul 15;21(14):4987. doi: 10.3390/ijms21144987. PMID: 32679729; PMCID: PMC7404275.
- Zhang, Y.; Ye, J. Mitochondrial Inhibitor as a New Class of Insulin Sensitizer. Acta Pharmaceutica Sinica B 2012, 2, 341–349, doi:10.1016/j.apsb.2012.06.010.
- Add reference for “have shown in clinical and preclinical studies to enhance lipid metabolism and insulin sensitivity”
Thank you for the helpful suggestion. We have now included relevant references to support this statement: "Additionally, PPAR agonists, such as pioglitazone (PPAR-γ) and the investigational GW501516 (PPAR-δ), have shown in clinical and preclinical studies to enhance lipid metabolism and insulin sensitivity”.
- Nagashima K, et al. Effects of the PPARgamma agonist pioglitazone on lipoprotein metabolism in patients with type 2 diabetes mellitus. J Clin Invest. 2005 May;115(5):1323-32. doi: 10.1172/JCI23219. Epub 2005 Apr 1. PMID: 15841215; PMCID: PMC1070635.
- Krämer DK, et al. Role of AMP kinase and PPARdelta in the regulation of lipid and glucose metabolism in human skeletal muscle. J Biol Chem. 2007 Jul 6;282(27):19313-20. doi: 10.1074/jbc.M702329200. Epub 2007 May 11. PMID: 17500064.
- Tanaka T, et al. Activation of peroxisome proliferator-activated receptor delta induces fatty acid beta-oxidation in skeletal muscle and attenuates metabolic syndrome. Proc Natl Acad Sci U S A. 2003 Dec 23;100(26):15924-9. doi: 10.1073/pnas.0306981100. Epub 2003 Dec 15. PMID: 14676330; PMCID: PMC307669.
- In conclusion, the authors mention that pharmacological mimetics can stimulate exercise-induced benefits in various diseases. However, in the texts, authors did not discuss how these agents can improve exercise performance to increase the exercise-mediated benefits in the aging group.
Thank you for this insightful comment. We have added a new paragraph in the revised manuscript (section 6) that specifically explains how pharmacological mimetics could enhance exercise performance and thus amplify exercise-mediated benefits in older adults. The new text reads as follows:
"In addition to their role as exercise mimetics, several of these pharmacological agents may also enhance exercise performance, thereby amplifying exercise-mediated benefits in older adults. For example, AMPK activators such as metformin and berberine improve mitochondrial efficiency and energy substrate utilization, which can reduce fatigue and increase endurance capacity. Similarly, SIRT1 activators like resveratrol and NAD⁺ precursors have been shown to enhance mitochondrial biogenesis and oxidative metabolism, leading to improved muscle function and recovery. Mitochondrial-derived peptides such as MOTS-c may further promote fatigue resistance and enhance muscle strength by activating key metabolic pathways and supporting mitochondrial health. These adaptations could help older adults tolerate and sustain regular physical activity, thus maximizing the combined benefits of exercise and pharmacological support”.
- Additionally, in Table 1, the authors did not provide an exercise protocol along with pharmacological agents that target or improve signaling molecules such as AMPK, PGC-1α, and Sir1. Authors should include the use of these agents with exercise protocols in the table to mimic these signaling molecules in different age-related diseases
Thank you for your valuable suggestion. We have now revised Table 1 to include exercise protocols (e.g., aerobic, resistance, or combined training regimens) alongside pharmacological agents that target or improve key signalling molecules such as AMPK, PGC-1α, and SIRT1. These additions provide a clearer framework for how exercise and pharmacological approaches can be integrated to optimize signalling and metabolic benefits in different age-related diseases.

Reviewer 2 Report
Comments and Suggestions for Authors
This is a well-written and comprehensive narrative review that brings together molecular, physiological, and therapeutic insights into muscle-adipose tissue crosstalk, particularly in the context of aging and metabolic flexibility. Rather than simply summarizing existing knowledge, the article takes a meaningful step forward by connecting basic mechanisms to potential therapeutic strategies, bridging the gap between bench and bedside in the field of aging and metabolism. It serves as a valuable follow-up to mechanistic reviews like Fang et al. (2022), adding a clear translational focus. While the review is well-organized and highly relevant for researchers and clinicians working in gerontology, endocrinology, and exercise science, a few sections would benefit from clearer phrasing and deeper critical analysis.
- The authors have referenced MOTS-c in several parts of the article. However, as an emerging therapeutic agent, MOTS-c still requires more preclinical data and a clearer discussion of its limitations. Authors should include this information and that would provide readers with a more balanced and informative context.
- Overall, this is a well-written review. However, in some sections, it leans more toward narrative storytelling and lacks a focused discussion on key limitations, conflicting findings, and barriers to clinical translation. Addressing these points would strengthen the review and enhance its value to the field.
- There are very few grammatical errors in this manuscript. E.g., “acute acute exercise” in line 88). Kindly recheck the entire text before resubmission.
- Authors have introduced Irisin as a myokine at the start of section 3.2 (Line 260). It is not required to introduce again in line 295.
- The conclusion does not acknowledge current limitations in biomarker stratification, interindividual variability, or long-term safety of exercise mimetics. Authors should add a brief cautionary note on translational gaps and need for large-scale trials.
- Authors are encouraged to clarify the distinction between PPAR-γ and PPAR-δ in metabolic flexibility context.
- Authors should provide appropriate references for usage of AAS and omega-3 to mitigate the chronic low-grade inflammation commonly associated with aging and metabolic dysfunction.
- Authors are requested to go through the 2022 review “Adipose–Muscle Crosstalk in Age-Related Metabolic Disorders” by Fang et al.
- Authors are suggested to include a short section outlining the current limitations and potential future research directions would round out the review nicely and give readers a clearer sense of what’s still needed in this area.
Author Response
Dear Respected Reviewer 2,
We would like to sincerely thank you for your careful and thoughtful evaluation of our manuscript. We greatly appreciate the time and effort dedicated to providing constructive feedback, which has helped improve the quality and clarity of the article.
Kindly find the revisions made to the manuscript in accordance with your suggestions, highlighted using the track changes feature.
This is a well-written and comprehensive narrative review that brings together molecular, physiological, and therapeutic insights into muscle-adipose tissue crosstalk, particularly in the context of aging and metabolic flexibility. Rather than simply summarizing existing knowledge, the article takes a meaningful step forward by connecting basic mechanisms to potential therapeutic strategies, bridging the gap between bench and bedside in the field of aging and metabolism. It serves as a valuable follow-up to mechanistic reviews like Fang et al. (2022), adding a clear translational focus. While the review is well-organized and highly relevant for researchers and clinicians working in gerontology, endocrinology, and exercise science, a few sections would benefit from clearer phrasing and deeper critical analysis.
- The authors have referenced MOTS-c in several parts of the article. However, as an emerging therapeutic agent, MOTS-c still requires more preclinical data and a clearer discussion of its limitations. Authors should include this information and that would provide readers with a more balanced and informative context.
We thank the reviewer for this insightful comment. In response, we have revised the relevant sections of the manuscript to include a more critical appraisal of MOTS-c as an emerging therapeutic agent. Specifically, we now acknowledge the limited preclinical evidence available, the need for standardized dosing and delivery protocols, and the lack of large-scale clinical trials in aging or metabolic disease populations. This updated discussion provides a more balanced view of the potential and current limitations of MOTS-c and has been added to Section 6 of the revised manuscript. The inserted paragraph is:
“Nevertheless, it is important to consider that the current evidence on MOTS-c remains predominantly preclinical [91]. In mouse models, MOTS-c administration has been shown to enhance physical performance across various age groups, including when treatment is initiated in older adulthood [92]. In humans, D’Souza et al. (2020) reported age-related declines in plasma MOTS-c levels, while muscle tissue levels paradoxically increased in healthy males—possibly reflecting shifts in muscle fiber type composition with aging [93]. However, clinical data are still limited. MOTS-c analogues, such as CB4211, are currently undergoing early Phase 1 trials, and key aspects including long-term safety, dosing regimens, pharmacokinetics, and therapeutic efficacy in aging or disease-specific populations have yet to be established [94].”
- Wan, W.; Zhang, L.; Lin, Y.; Rao, X.; Wang, X.; Hua, F.; Ying, J. Mitochondria-Derived Peptide MOTS-c: Effects and Mechanisms Related to Stress, Metabolism and Aging. J Transl Med 2023, 21, doi:10.1186/s12967-023-03885-2.
- Reynolds, J.C.; Lai, R.W.; Woodhead, J.S.T.; Joly, J.H.; Mitchell, C.J.; Cameron-Smith, D.; Lu, R.; Cohen, P.; Graham, N.A.; Benayoun, B.A.; et al. MOTS-c Is an Exercise-Induced Mitochondrial-Encoded Regulator of Age-Dependent Physical Decline and Muscle Homeostasis. Nat Commun 2021, 12, 470, doi:10.1038/s41467-020-20790-0.
- D’Souza, R.F.; Woodhead, J.S.T.; Hedges, C.P.; Zeng, N.; Wan, J.; Kumagai, H.; Lee, C.; Cohen, P.; Cameron-Smith, D.; Mitchell, C.J.; et al. Increased Expression of the Mitochondrial Derived Peptide, MOTS-c, in Skeletal Muscle of Healthy Aging Men Is Associated with Myofiber Composition. Aging 2020, 12, 5244–5258, doi:10.18632/aging.102944.
- Cognitive Vitality Reports® CB4211, 2021.
- Overall, this is a well-written review. However, in some sections, it leans more toward narrative storytelling and lacks a focused discussion on key limitations, conflicting findings, and barriers to clinical translation. Addressing these points would strengthen the review and enhance its value to the field.
We sincerely thank the reviewer for the constructive feedback and positive evaluation. In response, we have revised several sections of the manuscript to adopt a more analytical tone. We have incorporated a focused discussion on key limitations, including interspecies differences in preclinical models, the inconsistent outcomes of clinical trials involving exercise mimetics, and the challenges of translating molecular findings into real-world interventions. Additionally, we now explicitly address barriers to clinical translation such as biomarker heterogeneity, patient stratification difficulties, and the need for long-term safety data. These revisions are detailed in the newly added section titled “Limitations and Future Research Directions” (Section 8) and are briefly reiterated in the revised Conclusion to provide a balanced and forward-looking perspective.
- There are very few grammatical errors in this manuscript. E.g., “acute acute exercise” in line 88). Kindly recheck the entire text before resubmission.
We thank the reviewer for pointing this out. The typographical error “acute acute exercise” on line 88 has been corrected. In addition, we have carefully proofread the entire manuscript to ensure grammatical accuracy and clarity prior to resubmission.
- Authors have introduced Irisin as a myokine at the start of section 3.2 (Line 260). It is not required to introduce again in line 295.
Thank you for your suggestion, we removed myokine in line 295.
- The conclusion does not acknowledge current limitations in biomarker stratification, interindividual variability, or long-term safety of exercise mimetics. Authors should add a brief cautionary note on translational gaps and need for large-scale trials.
We thank the reviewer for this helpful suggestion. In response, we confirm that a dedicated section titled “Limitations and Future Research Directions” has already been included (Section 8), where we address the translational challenges, variability in individual responses, and the need for large-scale trials and long-term safety data on exercise mimetics. However, to reinforce these critical points, we have also added a brief cautionary statement to the Conclusion section, summarizing the most pressing limitations and emphasizing the importance of future clinical validation.
- Authors are encouraged to clarify the distinction between PPAR-γ and PPAR-δ in metabolic flexibility context.
We thank the reviewer for this insightful suggestion. In the revised version of the manuscript, we have included a dedicated paragraph (Section 6) clarifying the functional distinction between PPAR-γ and PPAR-δ in the context of metabolic flexibility. Specifically, we highlight that PPAR-γ primarily regulates insulin sensitivity and lipid storage in adipose tissue, whereas PPAR-δ promotes fatty acid oxidation and mitochondrial function in skeletal muscle. Relevant references have been added to support this clarification. The following paragraph was added:
“Both PPAR-γ and PPAR-δ activation may prove beneficial in maintaining metabolic flexibility during aging, as PPAR-γ, expressed in adipose tissue, enhances insulin sensitivity, while PPAR-δ, predominantly found in skeletal muscle, supports oxidative metabolism and promotes mitochondrial biogenesis.”
- Erol A. The Functions of PPARs in Aging and Longevity. PPAR Res. 2007;2007:39654. doi: 10.1155/2007/39654. PMID: 18317516; PMCID: PMC2254525.
- Tontonoz, P., Spiegelman, B.M. Fat and beyond: the diverse biology of PPARγ. Annu. Rev. Biochem. 2008, 77, 289–312. https://doi.org/10.1146/annurev.biochem.77.061307.091829
- Authors should provide appropriate references for usage of AAS and omega-3 to mitigate the chronic low-grade inflammation commonly associated with aging and metabolic dysfunction.
We thank the reviewer for this pertinent observation. In response, we have added appropriate references in the revised manuscript (Section 6, paragraph 3) to support the use of omega-3 polyunsaturated fatty acids and low-dose acetylsalicylic acid (AAS) as modulators of chronic low-grade inflammation associated with aging and metabolic dysfunction. Specifically, we cite clinical and preclinical studies demonstrating that omega-3s reduce systemic inflammation by suppressing NF-κB signaling and proinflammatory cytokine production, while low-dose AAS exerts anti-inflammatory and antiplatelet effects. The following references have been incorporated:
- Serhan, C.N., Levy, B.D. Resolvins in inflammation: emergence of the pro-resolving superfamily of mediators. J. Clin. Invest. 2018, 128(7), 2657–2669. https://doi.org/10.1172/JCI97943
- Bischoff-Ferrari, H.A., Gängler, S., Wieczorek, M. et al. Individual and additive effects of vitamin D, omega-3 and exercise on DNA methylation clocks of biological aging in older adults from the DO-HEALTH trial. Nat Aging 5, 376–385 (2025). https://doi.org/10.1038/s43587-024-00793-y
- Espinoza SE, Woods RL, Ekram ARMS, Ernst ME, Polekhina G, Wolfe R, Shah RC, Ward SA, Storey E, Nelson MR, Reid CM, Lockery JE, Orchard SG, Trevaks R, Fitzgerald SM, Stocks NP, Chan A, McNeil JJ, Murray AM, Newman AB, Ryan J. The Effect of Low-Dose Aspirin on Frailty Phenotype and Frailty Index in Community-Dwelling Older Adults in the ASPirin in Reducing Events in the Elderly Study. J Gerontol A Biol Sci Med Sci. 2022 Oct 6;77(10):2007-2014. doi: 10.1093/gerona/glab340. PMID: 34758073; PMCID: PMC9536436.
- Authors are requested to go through the 2022 review “Adipose–Muscle Crosstalk in Age-Related Metabolic Disorders” by Fang et al.
We appreciate the reviewer’s recommendation to consult the comprehensive 2022 review by Fang et al. on adipose–muscle crosstalk in age-related metabolic disorders. We incorporated key insights into the revised manuscript (Section 5), highlighting critical adipokines and myokines, such as adiponectin, spexin, irisin, and IL‑6, as central mediators of muscle‑fat communication in aging. We cite Fang et al. to reinforce our discussion on therapeutic strategies targeting adipo‑myokine signalling for metabolic flexibility and sarcopenic obesity prevention. The following paragraph and reference were added:
“Recent comprehensive reviews, notably by Fang et al. (2022), have synthesized evidence on the roles of specific adipokines (specifically adiponectin and spexin) and myokines (mainly irisin and IL‑6) in mediating adipose–muscle crosstalk that regulates energy homeostasis in age-related metabolic disorders. Disruption in these signalling pathways contributes to the pathogenesis of sarcopenic obesity, which Fang et al. describe as a convergence of high fat mass and declining muscle function with aging.”
Fang P, She Y, Yu M, Min W, Shang W, Zhang Z. Adipose-Muscle crosstalk in age-related metabolic disorders: The emerging roles of adipo-myokines. Ageing Res Rev. 2023 Feb; 84:101829. doi: 10.1016/j.arr.2022.101829.
- Authors are suggested to include a short section outlining the current limitations and potential future research directions would round out the review nicely and give readers a clearer sense of what’s still needed in this area.
We thank the reviewer for this thoughtful recommendation. In the revised manuscript, we have included a dedicated section titled “Limitations and Future Research Directions” (Section 8), which highlights key gaps in current knowledge and suggests priorities for future investigation. In this section, we discuss the challenges of interspecies extrapolation of findings from preclinical studies, the need for long-term clinical data on exercise mimetics, and the importance of developing stratified, biomarker-guided approaches tailored to aging populations. We hope this addition enhances the overall clarity and relevance of the review by offering a forward-looking perspective. The following paragraphs were added:
“Although increased attention has recently been given to studies focused on maintaining metabolic flexibility through physical activity in aging, and significant progress has been made in elucidating the molecular mediators of muscle–fat crosstalk, several limitations remain. Much of the current evidence is based on preclinical models or short-term clinical studies, which limits its extrapolation to older adults with multiple comorbidities. Furthermore, the long-term safety and efficacy of pharmacological “exercise mimetics”, such as AMPK activators, PPAR agonists, myostatin inhibitors, and mitochondrial peptides, require further validation in diverse, aging populations.
Moreover, interindividual variability in response to both exercise and pharmacological interventions is poorly understood and may be influenced by genetic background, microbiome composition, sex differences, and epigenetic factors. The translational gap between molecular findings and clinically applicable therapies remains substantial.
Future research should prioritize large-scale, longitudinal studies that incorporate biomarker-guided stratification, multi-omics approaches, and precision combinations of exercise and pharmacotherapy. Special attention should be given to evaluating interventions that concurrently target inflammation, mitochondrial function, and hormonal regulation. Additionally, clinical trials specifically designed for frail, sedentary, or mobility-impaired elderly individuals are needed to assess the therapeutic applicability of the proposed strategies.”

Round 2
Reviewer 1 Report
Comments and Suggestions for Authors
Authors have adequately improved their revised version. However, some corrections are still required to enhance the manuscript, especially since the authors did not provide the introduction. Therefore, authors must give an introduction rather than abruptly starting with headings or subheadings.
Minor corrections
In the table, why does the author mention Comorbidity/Feature? What does “Feature” mean here in this table?
In Figure 4, the author should mention acute and chronic exercise, rather than just mentioning acute and chronic.
Author Response
Dear Respected Reviewer 1,
We would like to sincerely thank you for your careful and thoughtful evaluation of our manuscript. We greatly appreciate the time and effort dedicated to providing constructive feedback, which has helped improve the quality and clarity of the article.
Kindly find the revisions made to the manuscript in accordance with your suggestions, highlighted using the track changes feature.
Authors have adequately improved their revised version. However, some corrections are still required to enhance the manuscript, especially since the authors did not provide the introduction. Therefore, authors must give an introduction rather than abruptly starting with headings or subheadings.
We sincerely thank the reviewer for the positive evaluation of our revised manuscript and for highlighting the absence of a formal Introduction section.
Accordingly, we have added an Introduction section at the beginning of the manuscript, prior to any subheadings. This section briefly outlines the significance of metabolic flexibility in aging, the molecular interplay between skeletal muscle and adipose tissue during exercise, and the rationale for exploring pharmacological strategies that mimic or enhance exercise-mediated signalling.
Please find the newly added Introduction section below:
“Aging is associated with progressive declines in metabolic flexibility, skeletal muscle function, and mitochondrial efficiency, which collectively contribute to the development of several pathologic conditions such as sarcopenia, insulin resistance, and cardiometabolic diseases [1]. Regular physical activity mitigates many of these age-related impairments by promoting favourable molecular adaptations in skeletal muscle and adipose tissue. Through the release of myokines and adipokines, exercise enables inter-tissue communi-cation that enhances lipid oxidation, glucose uptake, and mitochondrial biogenesis, thus supporting metabolic homeostasis and delaying functional decline [2–4].
Nonetheless, a significant proportion of older adults are unable to maintain regular physical activity due to frailty, multimorbidity, or physical limitations. This challenge has stimulated interest in developing pharmacological agents that mimic or potentiate the molecular benefits of exercise, commonly referred to as "exercise mimetics" (doi: 10.1016/j.cmet.2016.10.022). These agents often act on convergent signalling pathways such as AMPK–PGC-1α–SIRT1, PPARs, and inflammatory regulators like NF-κB to restore energy balance, reduce inflammation, and maintain muscle–fat metabolic communica-tion [5,6].”
Minor corrections
- In the table, why does the author mention Comorbidity/Feature? What does “Feature” mean here in this table?
Thank you for your valuable feedback regarding the terminology used in Table 1. We have replaced the column heading “Comorbidity/Feature” with “Clinical Condition or Metabolic Characteristic” to improve clarity and precision.
The revised term more accurately reflects the scope of the table, which includes both formally diagnosed comorbidities (e.g., cardiovascular disease, type 2 diabetes) and important physiological or metabolic traits (e.g., mitochondrial dysfunction, chronic inflammation, frailty) that may influence therapeutic decisions.
- In Figure 4, the author should mention acute and chronic exercise, rather than just mentioning acute and chronic.
Thank you for your observation regarding the labelling in Figure 4. We have revised the figure legend to explicitly refer to “acute and chronic physical exercise” rather than simply “acute and chronic.” This change ensures that the figure clearly communicates the intended comparison between short-term and long-term exercise-induced adaptations in skeletal muscle and adipose tissue.

Reviewer 2 Report
Comments and Suggestions for Authors
The revised manuscript reflects careful attention to the feedback provided during the initial review. Notable improvements include a more critical appraisal of emerging agents like MOTS-c, clearer differentiation between PPAR-γ and PPAR-δ, and thoughtful incorporation of relevant literature such as the Fang et al. (2022) review. The addition of references supporting anti-inflammatory strategies, along with the new section discussing limitations and future directions, has helped enhance the manuscript’s depth and translational relevance. Minor typographical issues and redundancies noted earlier have also been addressed. Overall, the revisions have strengthened the clarity, quality, and scope of the work.
Author Response
Many thanks to the reviewer for their positive comments
All our BW
Round 3
Reviewer 1 Report
Comments and Suggestions for Authors
In Table 1, why does the author include metabolic characteristics? Do the authors mean that frailty and renal impairment are metabolic characteristics?
Since my first revision, I suggested that the author should prepare the manuscript with careful attention. For example, the introduction was too brief, and the authors included a DOI number without following the journal's manuscript guidelines. Authors should elaborate the introduction section according to their concepts and follow the guidelines without indicating the DOI number.
Author Response
Dear Respected Reviewer 1,
We would like to sincerely thank you for your careful and thoughtful evaluation of our manuscript. We greatly appreciate the time and effort dedicated to providing constructive feedback, which has helped improve the quality and clarity of the article.
Kindly find the revisions made to the manuscript in accordance with your suggestions, highlighted using the track changes feature.
In Table 1, why does the author include metabolic characteristics? Do the authors mean that frailty and renal impairment are metabolic characteristics?
Thank you for your thoughtful question regarding the use of the term “metabolic characteristics” in Table 1. We appreciate the opportunity to clarify.
In the revised version, we used the broader term “Clinical Condition or Metabolic Characteristic” to encompass both formal diagnoses (e.g., type 2 diabetes, cardiovascular disease) and relevant physiological traits or syndromes that influence exercise capacity or metabolic flexibility, even if they are not strictly classified as metabolic disorders.
We agree that frailty and renal impairment are not traditionally categorized as metabolic characteristics per se. However, they are included in the table because they are clinically relevant states that significantly impact mitochondrial function, substrate utilization, inflammation, and energy homeostasis, processes that are central to the body's metabolic responses to exercise and pharmacological agents.
To avoid confusion, we have now clarified this in the table caption and ensured that the column heading emphasizes the inclusion of both diagnosed conditions and clinically relevant traits that affect therapeutic decisions in older adults.
Updated Column Heading: Clinical Condition or Functional/Metabolic Trait
Updated Table Caption: Recommended pharmacological agents and exercise strategies for aging individuals, stratified by diagnosed clinical conditions (T2DM, cardiovascular disease) and relevant functional or metabolic traits (frailty, chronic inflammation, renal impairment) that influence metabolic flexibility, exercise tolerance, or treatment safety in older adults.
- Since my first revision, I suggested that the author should prepare the manuscript with careful attention. For example, the introduction was too brief, and the authors included a DOI number without following the journal's manuscript guidelines. Authors should elaborate the introduction section according to their concepts and follow the guidelines without indicating the DOI number.
We thank the reviewer for the continued constructive feedback. In response to the earlier recommendation, we have thoroughly revised the Introduction section to provide a more comprehensive and conceptually aligned overview of the manuscript’s objectives and scope. The expanded introduction now includes recent literature (2020 and onward) to support the rationale and context, particularly regarding aging, metabolic flexibility, muscle–fat crosstalk, and the potential role of exercise mimetics.
Additionally, we have carefully revised the manuscript to ensure full compliance with the journal’s formatting requirements. The DOI number have been removed from the in-text references and the reference list, as per the journal’s guidelines.
We appreciate the reviewer’s guidance, which has significantly improved the structure and clarity of the manuscript.

Round 4
Reviewer 1 Report
Comments and Suggestions for Authors
The authors followed all my suggestions, and the manuscript can be accepted in its present form.